# Syncytial germline architecture is actively maintained by contraction of an internal actomyosin corset

Agarwal Priti[1], Hui Ting Ong[1], Yusuke Toyama[1,2], Anup Padmanabhan [1], Sabyasachi Dasgupta[1,3], Matej Krajnc[4,5] & Ronen Zaidel-Bar[1,6]

Syncytial architecture is an evolutionarily-conserved feature of the germline of many species and plays a crucial role in their fertility. However, the mechanism supporting syncytial organization is largely unknown. Here, we identify a corset-like actomyosin structure within the syncytial germline of *Caenorhabditis elegans*, surrounding the common rachis. Using laser microsurgery, we demonstrate that actomyosin contractility within this structure generates tension both in the plane of the rachis surface and perpendicular to it, opposing membrane tension. Genetic and pharmacological perturbations, as well as mathematical modeling, reveal a balance of forces within the gonad and show how changing the tension within the actomyosin corset impinges on syncytial germline structure, leading, in extreme cases, to sterility. Thus, our work highlights a unique tissue-level cytoskeletal structure, and explains the critical role of actomyosin contractility in the preservation of a functional germline.

[1] Mechanobiology Institute, National University of Singapore, Singapore 117411, Singapore. [2] Department of Biological Sciences, National University of Singapore, Singapore 117543, Singapore. [3] Department of Physics, University of Toronto, 60 St. George St., Toronto, ON M5S 1A7, Canada. [4] Lewis-Sigler Institute for Integrative Genomics, Princeton University, Washington Road, Princeton 08540, USA. [5] Jožef Stefan Institute, Jamova 39, SI-1000 Ljubljana, Slovenia. [6] Department of Cell and Developmental Biology, Faculty of Medicine, Tel Aviv University, POB 39040, Tel Aviv 69978, Israel. Correspondence and requests for materials should be addressed to R.Z.-B. (email: zaidelbar@tauex.tau.ac.il)

Cell shape and tissue architecture play an important role in regulating organ function and homeostasis. One unique and well-conserved tissue architecture is exhibited by the germline syncytium, wherein multiple nuclei share a common cytoplasm[1–3]. An excellent model to study germline syncytial architecture is the nematode *Caenorhabditis elegans*. The *C. elegans* hermaphrodite has a single gonad with two U-shaped tubular arms each containing approximately one thousand germ cells in a syncytium (Fig. 1a)[4]. The germ cells are arranged peripherally within the gonad and are only partially enclosed with plasma membrane, sharing a common central cytoplasm known as rachis. The syncytial structure of the germline arises progressively from larval stages to the adult[5]. Intercellular bridges, known as rachis bridges, connecting the germ cells to the rachis (Fig. 1b, c), are reminiscent of the ring canals of nurse cells in the *Drosophila* ovary[2,6,7] and intercellular bridges observed in mammals[1,8]. A cytoplasmic flow through the rachis bridges of the mitotic and meiotic germ cells, present at the distal end of the gonad, provides the cytoplasmic material to the growing proximal germ cells which eventually enlarge and cellularize to form oocytes[9,10]. These rachis bridges are enriched in some actomyosin regulators[11–14]. Recently, several studies have revealed the role of actomyosin regulators in the stabilization of rachis bridges[5,15–18], however, their function in maintenance of syncytial tissue organization has not been explored.

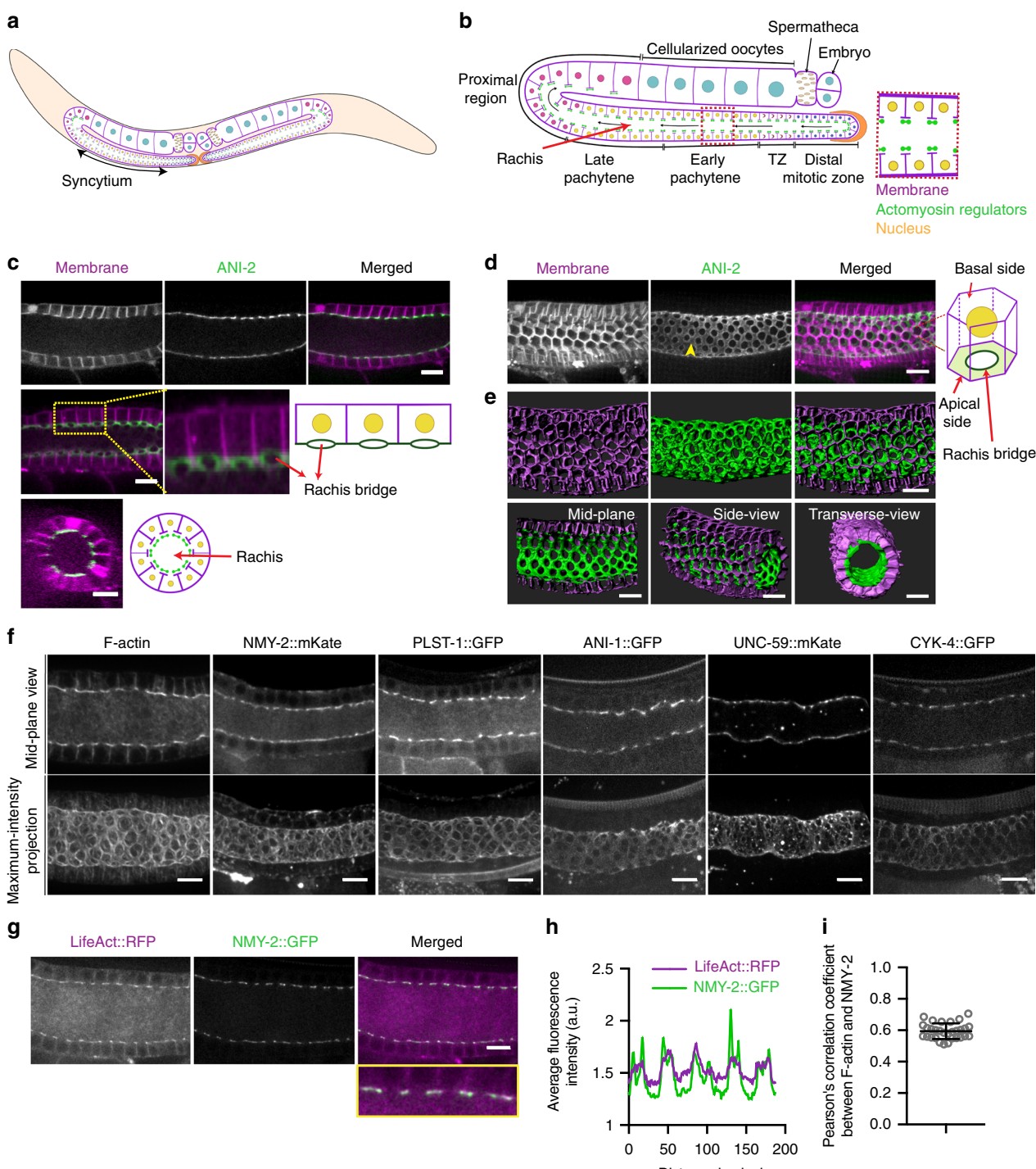

**Fig. 1** Actomyosin regulators form an inner corset enclosing the rachis of the *C. elegans* germline. **a** Schematic overview of the entire *C. elegans* hermaphrodite reproductive system. **b** Detailed schematic of a mid-plane view of a single arm of the *C. elegans* gonad. The germ cells enter mitosis at the distal end and progress through different stages of meiotic development to form mature oocytes at the proximal end. These germ cells are present in a syncytium, with openings into a central rachis. The T-region of germ cell membranes (magenta) in enriched in actomyosin regulators (green). Blue—mitotic nucleus, crescent shaped nucleus—transition zone nucleus, yellow—early meiotic nuclei, pink—diplotene stage nuclei, and cyan—diakinetic nuclei. **c** Top panel shows confocal images of a mid-plane view of the early meiotic region (transition and early pachytene) of the germline expressing mCherry::PH (magenta) and GFP::ANI-2 (green). Middle panel—projection of three consecutive slices showing the opening of germ cells into the rachis. Inset—magnified view and corresponding schematic explanation. Bottom panel—orthogonal view of the gonad and a schematic presentation. **d** Maximum-intensity projection of the early meiotic region of the germline expressing mCherry::PH (magenta) and GFP::ANI-2 (green). The germ cells have a hexagonal base and circular opening at the apical end. GFP::ANI-2 enriched outside the holes (yellow arrow heads) forming a tubular structure around the rachis. **e** Imaris reconstruction of the gonad showing three-dimensional views from different angles. **f** Mid-plane and maximum-intensity projected views of the distal region of the gonads expressing different actomyosin regulators. **g** Confocal image of the germline co-expressing LifeAct::RFP and NMY-2::GFP. Merged image and the inset show their co-localization at the rachis surface. **h** Intensity line profile of the proteins obtained along the rachis surface show in the inset of (**g**). **i** Pearson's coefficient of correlation (PCC) calculated between LifeAct::RFP and NMY-2::GFP by drawing a seven pixel line at the rachis bridge of the distal end of 30 different gonads. Scale bar, 10 μm

Owing to presence of contractility regulators at the rachis bridge, we explored the role of actomyosin contractility for the maintenance of syncytial tissue organization. Using three-dimensional image analysis, we show that the actomyosin machinery is not only enriched at rachis bridges but is also present between bridges forming a tissue-level actomyosin-rich structure, surrounding the rachis, akin to an inner corset. Laser microsurgery combined with time-lapse imaging show that this actomyosin corset is under tension, which depends on myosin activity. Furthermore, direction-specific laser incisions, genetic, and drug manipulations of several actomyosin regulators combined with quantitative image analysis show the effect of altered tension within the actomyosin corset on the structure of the syncytial germline and reveal the presence of two-directional contractile forces originating from the actomyosin corset to maintain germline architecture. Finally, we develop a mathematical model simulating the balance of forces within the gonad and show that contraction of the apical actomyosin corset can modulate the syncytial geometry in a manner consistent with our experimental results. Taken together, our study unveils how mechanical forces generated by an inner corset-like contractile cytoskeletal network maintain the structure of the *C. elegans* syncytial germline.

## Results

**A stable actomyosin corset exists within the *C. elegans* gonad.** To better understand the syncytial organization of the *C. elegans* germline, we employed live fluorescence confocal microscopy of the early meiotic region of a gonad co-expressing a membrane marker tagged to mCherry (mCherry::PH) and the adaptor protein anillin-2 fused to GFP (GFP::ANI-2). ANI-2 was previously shown to be localized to the rachis surface, enriched at intercellular bridges and required for their stability[4,5]. Consistent with those reports, single medial plane views showed the presence of high concentration of ANI-2 at the top of T-shaped radial partitions composed of the germ cell membranes surrounding the central rachis (Fig. 1c, upper row). Projection of three consecutive slices revealed the openings of germ cells into the rachis and ANI-2 localization around the rachis bridges (Fig. 1c, middle row). An orthogonal view of an early pachytene region of the gonad (~80–100 μm from the distal end) revealed the central rachis to be surrounded by 10–12 germ cells (Fig. 1c, bottom row). Maximum-intensity projections of the entire Z-stack and three-dimensional surface rendering highlights that GFP::ANI-2 forms a tubular structure within the gonad, separating the central rachis from the germ cells, with holes that correspond to the opening of each germ cell (Fig. 1d–e, Supplementary Fig. 1a and Supplementary Movie 1). A similar distribution pattern was observed for

F-actin, non-muscle myosin II (NMY-2), and several other actomyosin machinery proteins[19]: actin-bundling protein PLST-1, canonical anillin ANI-1, septin UNC-59, and RhoGAP protein CYK-4[11,13,14,20] (Fig. 1f). Importantly, a strong co-localization was observed between F-actin (marked by LifeAct::RFP) and NMY-2::GFP at the rachis (Pearson's correlation coefficient = +0.59 ± 0.05, mean ± s.d., number of gonads, $N = 30$) (Fig. 1g–i).

We used fluorescence recovery after photobleaching (FRAP) analysis to examine the dynamics of three fluorescently tagged endogenous actomyosin regulators within the actomyosin tube: GFP::ANI-2, NMY-2::mKate, and PLST-1::GFP. Interestingly, each of these proteins displayed different recovery kinetics: PLST-1 recovered the fastest ($t_{1/2} = 37 \pm 16$ s, $N = 10$) and had the largest mobile fraction ($79.5 \pm 1\%$, $N = 10$); NMY-2 recovered slower ($t_{1/2} = 104 \pm 48$ s, $N = 10$) and had approximately equal fractions of mobile and immobile pools ($53.3 \pm 1.1\%$ mobile, $N = 10$); and ANI-2 was the slowest to recover ($t_{1/2} = 129 \pm 54$ s, $N = 10$) and was mostly immobile ($29 \pm 5\%$ mobile, $N = 10$) (Supplementary Fig. 1b–e and Supplementary Movie 2). Compared with myosin dynamics measured in the cortex and cytokinetic ring in *C. elegans* embryos[21,22], these values indicate a much more stable structure, with low turnover and large immobile pools.

**Actomyosin corset is under myosin-dependent tension.** To test whether the syncytial actomyosin tube is under contractile tension, we performed laser ablations combined with time-lapse imaging. First, a ~9 μm line incision was performed at the apical region of the germ cells, labeled with GFP::ANI-2, in a distal to proximal direction (Fig. 2a, b). The surrounding membranes exhibited a rapid displacement with an initial recoil velocity of $1.12 \pm 0.28$ μm/s, $N = 24$ (Fig. 2b–d). Depletion of NMY-2 reduced the initial recoil velocity ($0.47 \pm 0.10$ μm/s, $N = 19$) by ~2.3 fold compared to the control, suggesting that most if not all the tension in the germline is myosin-dependent (Fig. 2b–d and Supplementary Movie 3). Consistent with a previous report of higher levels of phosphorylated myosin in the distal region of the gonad compared with the proximal region[23], point ablations at the rachis bridge in the distal end showed higher displacement than at the proximal end (Supplementary Fig. 1f–h and Supplementary Movie 4).

To further examine the directionality of tension in the distal region, we performed point ablations and linear ablations perpendicular to the distal to proximal direction at the rachis surface (Fig. 2a). Point ablations (between rachis bridges) showed uniform radial opening of the rachis surface, with an initial recoil velocity of $1.12 \pm 0.24$ μm/s, $N = 15$ (Fig. 2e, h, i, Supplementary Movie 5). Similar to ablations made along the distal to proximal

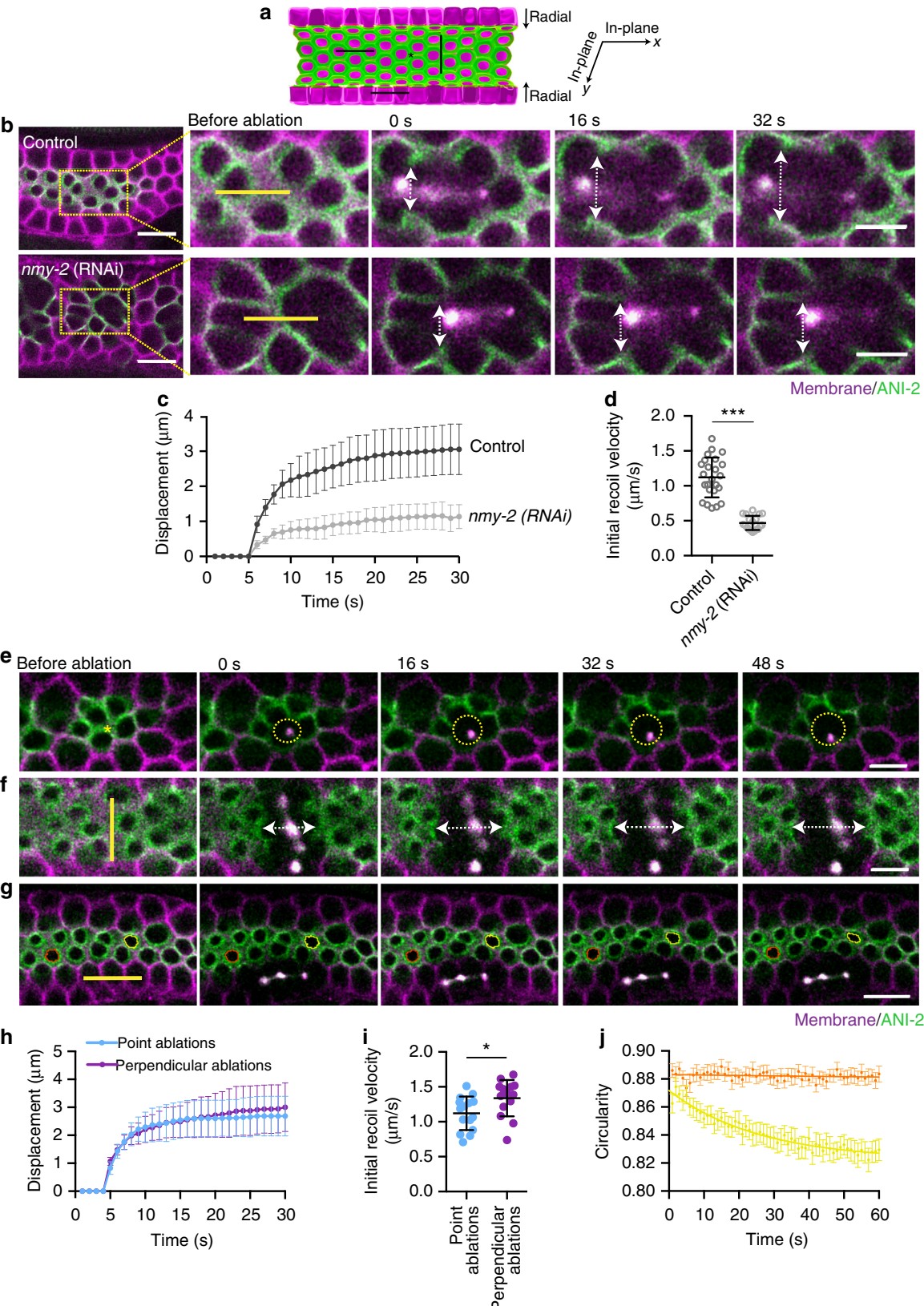

axis of the gonad (Fig. 2b–d), perpendicular ablations showed a rapid recoiling of the nearby surface with an initial recoil velocity of 1.33 ± 0.26 µm/s, $N = 14$ (Fig. 2f, h, i and Supplementary Movie 5). These ablation experiments showed that there is uniform tension acting in the plane of the rachis surface (Fig. 2a).

To test the presence of any radial force acting on the rachis surface, we ablated germ cell membranes (orthogonal to the rachis surface) and observed changes in the shape of rachis bridges of the neighboring cells with time (Fig. 2a, g). Ablating orthogonal membranes caused a gradual displacement of the

**Fig. 2** Actomyosin-enriched inner corset is under tension. **a** A schematic of half of the gonad showing where laser incisions were performed in **b**, **e**, **f**, and **g**. Green color represents actomyosin regulators enriched region while magenta color represents germ cell membrane. Asterisk shows the site of point incision and black lines shows the site of line incision along the distal to proximal axis, perpendicular line incision at the rachis surface, and ablation at the germ cell membrane. **b** Time-lapse images of the recoiling of germ cell membrane after laser incision at the rachis of control and *nmy-2* (RNAi) germlines. An incision was made in the direction of the distal to proximal axis (yellow line ~9 μm). White double-headed arrow shows the displacement at indicated time point. **c** Displacement of the adjacent membranes over time after ablation in control ($N = 24$) versus *nmy-2* (RNAi) ($N = 19$) germlines. **d** Quantification of initial recoil velocity after laser incision at the rachis in control ($N = 24$) versus *nmy-2* (RNAi) ($N = 19$) germlines. **e** Time-lapse images of the recoiling of rachis surface after point ablation. Dotted circle represents isotropic displacement of the nearby surface over time. **f** Time-lapse images of the recoiling of rachis surface after a line incision made perpendicular to the direction of the distal to proximal axis (yellow line ~9 μm). White double-headed arrow shows displacement at indicated time points. **g** Time-lapse images of shape deformation of the rachis bridges after laser incision at the germ cell membranes (orange—rachis bridge away from the incision site; yellow—rachis bridge near to the incision site). **h** Displacement of the adjacent rachis surface over time after point ($N = 15$) and perpendicular line incision ($N = 15$). **i** Quantification of initial recoil velocity after point ($N = 15$) and perpendicular line incision ($N = 14$). **j** Change in the circularity of rachis bridge over time. Error bar denotes standard deviation (s.d.), data represents mean ± s.d. (*$p < 0.05$, ***$p < 0.0001$) of three independent experiments. Statistical analysis was done using Mann–Whitney *U*-test, $N =$ number of gonads analyzed and scale bar represents 10 μm. Scale bar in the magnified images of (**a**) represents 5 μm

---

adjacent cells with a significant change in the circularity of rachis bridges, while rachis bridges of cells at a distance from the ablated region remained unaltered (Fig. 2g, j and Supplementary Movie 6). This indicates the presence of an inward radial force at the rachis surface carefully balanced by membrane tension.

Together, these results suggest that the inner tube of actomyosin machinery surrounding the rachis is a contractile structure with an isotropic distribution of tension at the rachis surface and a radial inward force opposing membrane tension.

**Contractile corset maintains the syncytial architecture.** Next, we sought to observe the effects of decreasing the level of actomyosin contractility on syncytial germline structure. Contractile forces are generated predominantly by the motor activity of myosin on actin filaments[24,25]. Therefore, actomyosin forces can be reduced either by reducing F-actin levels or by inhibiting myosin activity. We found the formin CYK-1, an endogenous actin nucleation and elongation factor, to localize prominently at the rachis envelope (Fig. 3a), consistent with a previous immunostaining result[12,26]. When worms carrying the temperature-sensitive allele *cyk-1(or596)* are grown at the semi-permissive temperature of 20 °C, CYK-1 activity is partially abrogated, leading to ~50% reduction in the level of F-actin at the rachis, as measured from phalloidin staining (Fig. 3b, c). Under these conditions, we observed pronounced structural defects in the early meiotic region of the germline (transition zone and early pachytene). Specifically, in comparison with the control, the height of germ cells was significantly reduced, while the perimeter and diameter of rachis bridges significantly increased (Fig. 3d–g). Stronger depletion of CYK-1, achieved by performing *cyk-1*(RNAi) on *cyk-1(or596)* worms, resulted in failure of germ cell cellularization and sterility (Supplementary Fig. 2a).

We examined the effect of loss of non-muscle myosin-II activity on germline architecture by depleting either NMY-1 or NMY-2, the two non-muscle myosin II isoforms, by RNAi. Depletion of NMY-1 did not result in any visible abnormality in the structure of the syncytial germline, but displayed abnormal oocytes and spermatheca contractility defects with embryos trapped in the spermatheca (Supplementary Fig. 2b) as reported previously[27]. Consistent with this phenotype, and contrary to a recent report[28], endogenous NMY-1 tagged with mKate was expressed strongly in the sheath cells and spermatheca, but was absent from the germline (Supplementary Fig. 2c).

Depleting NMY-2 by RNAi from the L3 stage resulted in sterile worms with complete loss of apical and lateral membranes of germ cells (Supplementary Fig. 2a). This strong phenotype precluded us from testing the balance of forces within the

syncytial germline. Therefore, we sought partial depletion of NMY-2 activity. Starting the *nmy-2*(RNAi) at the L4 stage resulted in an expansion of rachis diameter, with germ cell height ~1.5 times shorter than control and the perimeter of the germ cell apical openings ~two times larger than control, similar to the phenotype of *cyk-1(or596)* mutants at 20 °C (Fig. 3d–g). As another way to inhibit myosin activity, we depleted *let-502*, the Rho kinase that regulates myosin light chain phosphorylation and hence myosin activity during embryogenesis and in the somatic gonad[29–32]. Importantly, *let-502*(RNAi) phenocopied *nmy-2* (RNAi), suggesting that myosin motor activity is required for the structural integrity of germ cells (Fig. 3d–g).

The aforementioned genetic perturbations required several hours to manifest their phenotype. To achieve an acute inhibition of actomyosin contractility specifically within the germline, we performed pharmacological inhibition. Latrunculin A (5 μM), an inhibitor of actin-polymerization or ML-7 (250 μM), a myosin light-chain kinase inhibitor, were injected along with a DNA dye directly into the gonads of worms expressing GFP::ANI-2 and a membrane marker. Imaging the germline shortly after injection revealed a rapid (~1–2 min after Latrunculin A and ~40 min after ML-7 injection) and irreversible collapse of germ cell membranes[9], as observed for complete loss of function of CYK-1 and NMY-2 (Fig. 3h, i). Injection of DMSO alone did not perturb the germline structure.

To increase contractility within the germline, we depleted by RNAi negative regulators of myosin activity. MEL-11, a regulatory subunit of myosin phosphatase[30] is expressed within the germline and its depletion results in a remarkable increase in the level of phosphorylated myosin throughout the gonad[23]. In contrast to the phenotype observed by loss of F-actin or myosin activity, *mel-11*(RNAi) caused a small but significant increase in the height of germ cells (~1.13 fold) (Fig. 4a, b), a decrease in the perimeter of germ cell apical openings (~1.16 fold) (Fig. 4a, c), and a reduction in rachis diameter (Fig. 4a, d). GCK-1, germinal serine-threonine kinase and its co-factor, CCM-3 localize specifically on the rachis bridges, where they inhibit NMY-2 activity[15,16]. Knockdown of GCK-1 and CCM-3 resulted in gonads with taller germ cells, a narrower rachis, and smaller rachis bridges (Fig. 4a–d).

Together, our laser microsurgery and genetic perturbation results suggest that actomyosin contractility at the surface of the rachis generates forces in two orthogonal directions: in the plane of the rachis surface it acts to close the germ cell apical openings (rachis bridges) and perpendicular to the rachis surface it acts to pull the germ cell membranes towards the rachis center. At homeostasis, actomyosin contractility is carefully regulated to balance the membrane tension for the maintenance of

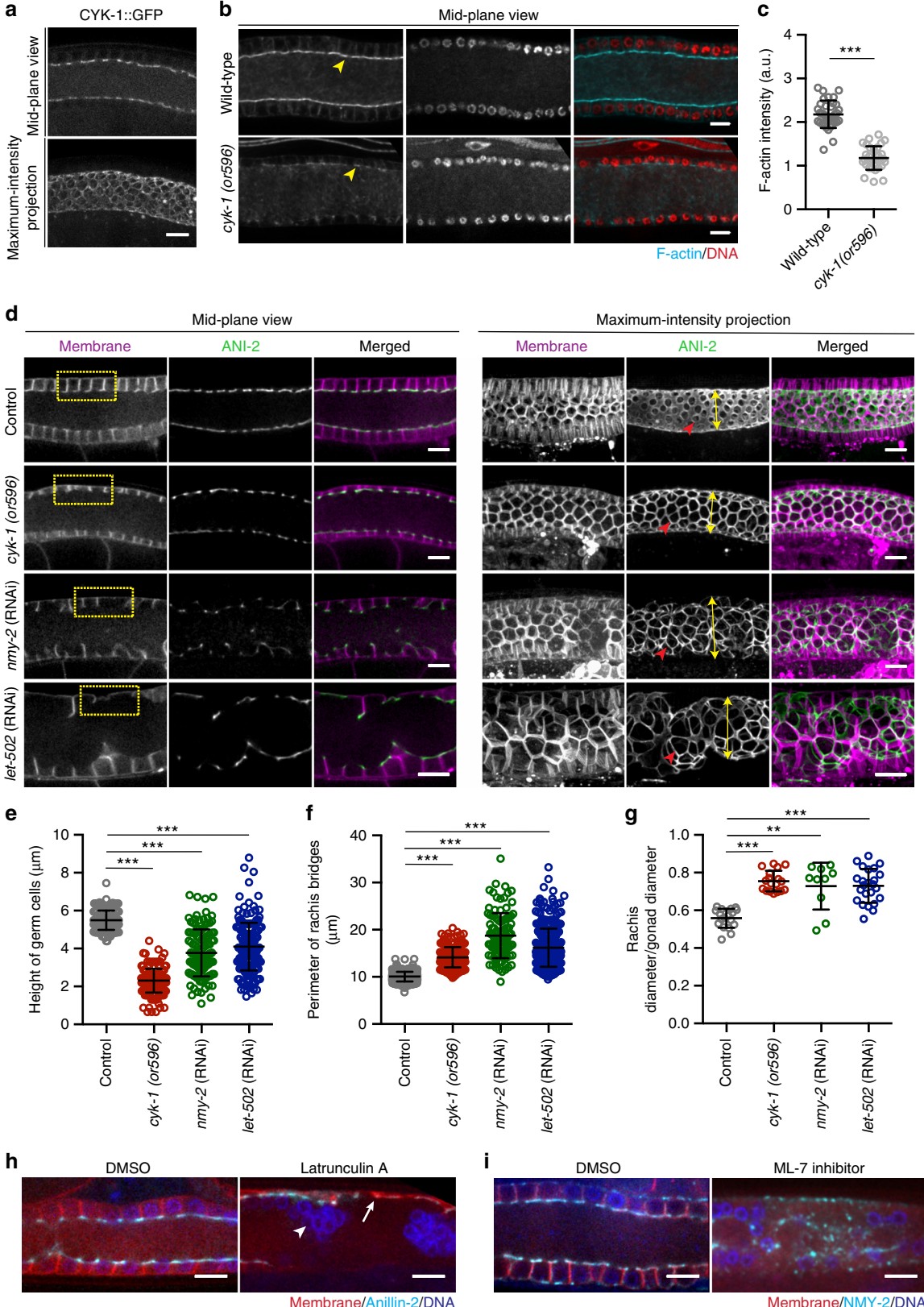

appropriate height of germ cells and proper openings of the cells into the rachis.

**Actomyosin contractility regulates cytoplasmic streaming.** Excessive contractility led to the formation of small and non-

functional oocytes[5,15,16] (Supplementary Fig. 3). Since oocyte growth is dependent on cytoplasmic streaming in the syncytial gonad[9,10], we examined the effect of modulating actomyosin contractility on cytoplasmic streaming. To this end, we acquired high speed differential intensity contrast images of control and

**Fig. 3** Loss of contractility results in shortening of germ cell membranes, increased intercellular bridge perimeter, and a wider rachis. **a** Confocal image of GFP labeled endogenous formin, CYK-1. **b** Phalloidin (cyan) and DAPI (red) stained gonads of wild-type and temperature-sensitive *cyk-1(or596)* mutants grown at semi-permissive temperature (20 °C). Yellow arrow heads show actin level in wild-type versus *cyk-1(or596)* mutants. **c** Quantification of actin intensity at rachis bridge in wild-type ($N = 35$) versus *cyk-1(or596)* mutants ($N = 32$). **d** Mid-plane and maximum-intensity projection views of the germline, expressing mCherry::PH (magenta) and GFP::ANI-2 (green). Reduced contractility was achieved by using *cyk-1(or596)* grown at 20 °C, or knockdown of non-muscle myosin-II (NMY-2), or Rho-binding Ser/Thr kinase, LET-502. Yellow rectangular boxes emphasize the height of germ cells. Red arrowheads highlight the difference in the perimeter of rachis bridge. Yellow double-headed arrows show the increase in rachis diameter. **e** Quantification of the height of germ cells in early meiotic region of the gonads of control ($n = 203$ membranes, $N = 20$), *cyk-1(or596)* at 20 °C ($n = 200$ membranes, $N = 20$), *nmy-2*(RNAi) ($n = 120$ membranes, $N = 10$) and *let-502*(RNAi) ($n = 200$ membranes, $N = 20$). **f** Quantification of the perimeter of rachis bridges in early meiotic region of the gonads of control ($n = 250$ bridges, $N = 23$), *cyk-1(or596)* at 20 °C ($n = 215$ bridges, $N = 18$), *nmy-2*(RNAi) ($n = 105$ bridges, $N = 16$), and *let-502*(RNAi) ($n = 250$ bridges, $N = 22$). **g** Quantification of rachis/gonad diameter in early meiotic region of the gonads of control ($N = 18$), *cyk-1(or596)* at 20 °C ($N = 18$), *nmy-2*(RNAi) ($N = 10$), and *let-502* (RNAi) ($N = 24$). **h** Confocal image of the germline expressing mCherry::PH;GFP::ANI-2 treated with DMSO (control) ($N = 5$) and Latrunculin A, 5 µM ($N = 5$). White arrow represents collapse of membranes and white arrowhead show clumped nuclei. **i** Confocal image of the germline expressing PH::GFP; NMY-2::mKate injected with DMSO (control) ($N = 9$) and myosin light-chain kinase inhibitor (ML-7), 250 µM for 40 min ($N = 6$). A DNA marker (blue), Hoechst dye, was used as control for injection. Data represents mean ± s.d. of three independent experiments. Statistical analysis was done using Mann–Whitney *U*-test (**c**) and one-way ANOVA test (**e**–**g**) (**$p < 0.005$, ***$p < 0.0001$). $N =$ number of gonads analyzed. Scale bar, 10 µm

perturbed germlines, and employed particle image velocimetry to obtain the average velocity (along the gonad long axis) and speed of cytoplasmic particles in the distal region of the gonad. The average speed and velocity of *cyk-1(or596)* mutants at 20 °C ($1.35 ± 0.90$ µm/min; $4.06 ± 1.20$ µm/min, respectively, $N = 22$) and *nmy-2*(RNAi) on L4 stage worms ($1.78 ± 0.81$ µm/min; $3.70 ± 1.09$ µm/min, $N = 10$, respectively) were not significantly different from the wild-type ($1.50 ± 0.50$ µm/min; $4.71 ± 1.63$ µm/ min, $N = 17$, respectively) (Fig. 5a–c and Supplementary Movie 7). However, more thorough removal of CYK-1, NMY-2, and knockdown of MEL-11 caused a significant decrease in the average velocity (along the gonad long axis) and speed of the particles (*cyk-1*(RNAi) on *cyk-1(or596)*: $0.21 ± 0.64$ µm/min; $2.63 ± 1.70$ µm/min, $N = 11$; *nmy-2*(RNAi) on L3 stage worms: $0.60 ± 0.70$ µm/min; $3.03 ± 1.67$ µm/min, $N = 10$; *mel-11*(RNAi): $0.45 ± 0.80$ µm/min; $3.09 ± 1.26$ µm/min, $N = 19$, respectively) (Fig. 5a–c and Supplementary Movie 8). These results show that actomyosin contractility is essential for cytoplasmic streaming, as suggested previously[9]. However, partial depletion of the acto-myosin regulators, which was sufficient to affect germline structure, did not affect the cytoplasmic flow. This demonstrates that the changes in germline architecture upon actomyosin regulator perturbations are not secondary to defects in cytoplasmic streaming.

**PLST-1, an actin-crosslinker, smoothens tension distribution**. In all experiments described thus far, actomyosin contractility was either increased or decreased uniformly across the entire gonad. While investigating the role of actomyosin regulators in the germline, we discovered that in a loss of function mutant of the actin cross-linker PLST-1 the distribution of F-actin and NMY-2 is highly non-uniform within a given syncytial germline (Fig. 6a–d). PLST-1, which we showed enhances actomyosin network connectivity in the zygotic cortex[20], also localized in and around rachis bridges, as seen for other contractome[19] proteins (Fig. 1f). In the absence of PLST-1, F-actin organization in rachis bridges, revealed by phalloidin staining, was dramatically altered, with apical openings varying in size and shape instead of the uniform round shape observed in controls (Fig. 6a). Although the total amount of myosin-II was unaffected in the absence of PLST-1, its distribution pattern was markedly variable compared to the wild-type gonads, with myosin levels being low in some regions and forming aggregates in other regions (Fig. 6b, c). Quantifica-tion of the coefficient of variation of myosin intensity along the rachis bridge confirmed the higher variability in *plst-1(tm4255)* mutants ($0.50 ± 0.07$, $N = 25$) as compared to the control ($0.32 ±$

$0.04$, $N = 31$) (Fig. 6d). Consistent with the high variability of myosin levels, laser ablations at the rachis of *plst-1(tm4255)* mutants revealed increased variability in tissue tension (Fig. 6e). These observations indicate that PLST-1 is required for the smooth distribution of tension within the germline, and gave us the opportunity to further investigate the relationship between actomyosin contractility and germline architecture, within the same worm. Compared to the control germline, *plst-1(tm4255)* mutants had a wider distribution of germ cell height and rachis bridge perimeter (Fig. 6f–h). Importantly, there was a strong positive correlation between local NMY-2::mKate intensity at the rachis and germ cell height in the same region (Fig. 6i, j). No such correlation was observed between the intensity of PH::GFP and height of the germ cell membranes (Fig. 6j). Altogether, these results indicate that PLST-1 is essential for maintaining a smooth distribution of actomyosin along the rachis, and the level of contractility determines rachis diameter, germ cell height, and the size of germ cell openings/rachis bridges.

**Numerical model supports the role of apical contractility**. To formulate how contractility of an actomyosin network at the apex of germ cells could regulate the germline architecture in the way we observed, we performed in silico modeling of the germline based on a three-dimensional (3D) vertex model[33–36]. The gonadal tube was modeled as a cylindrical epithelial sheet of closely packed cells with hexagonal bases and apices, defined by vertices, which move in response to mechanical forces, driving the system towards the minimum of the potential energy *w* (Fig. 7a and Supplementary Fig. 4a). A significant mechanical contribution arises from the contractile apical actomyosin ring, which we incorporated in the model by including apical con-tractility ($\alpha$) acting along the perimeter of the apical cell side[37,38]. Cell–cell adhesion and cortical tension that act over the lateral cell surfaces were described by an effective lateral surface tension[39] (set to unity). Furthermore, cells are assumed to have a preferred (unit) volume, corresponding to the average volume of cells in the wild-type condition (calculated by using experimental values of gonad diameter, rachis diameter, and number of germ cells within a defined region) and compressibility ($\psi$), accounting for their deformable interior[40] (Supplementary Fig. 4b). This description is used to effectively capture cell-volume ($v$) changes due to com-pressibility of intracellular cytoplasmic fluid, cortical network present within the cell as well as passive exchange of material with the environment. We did not explicitly model any osmotic or active transport due to fluxes of solutes or fluid[41]. The basal side of the tissue was assumed to be fixed. This assumption was based

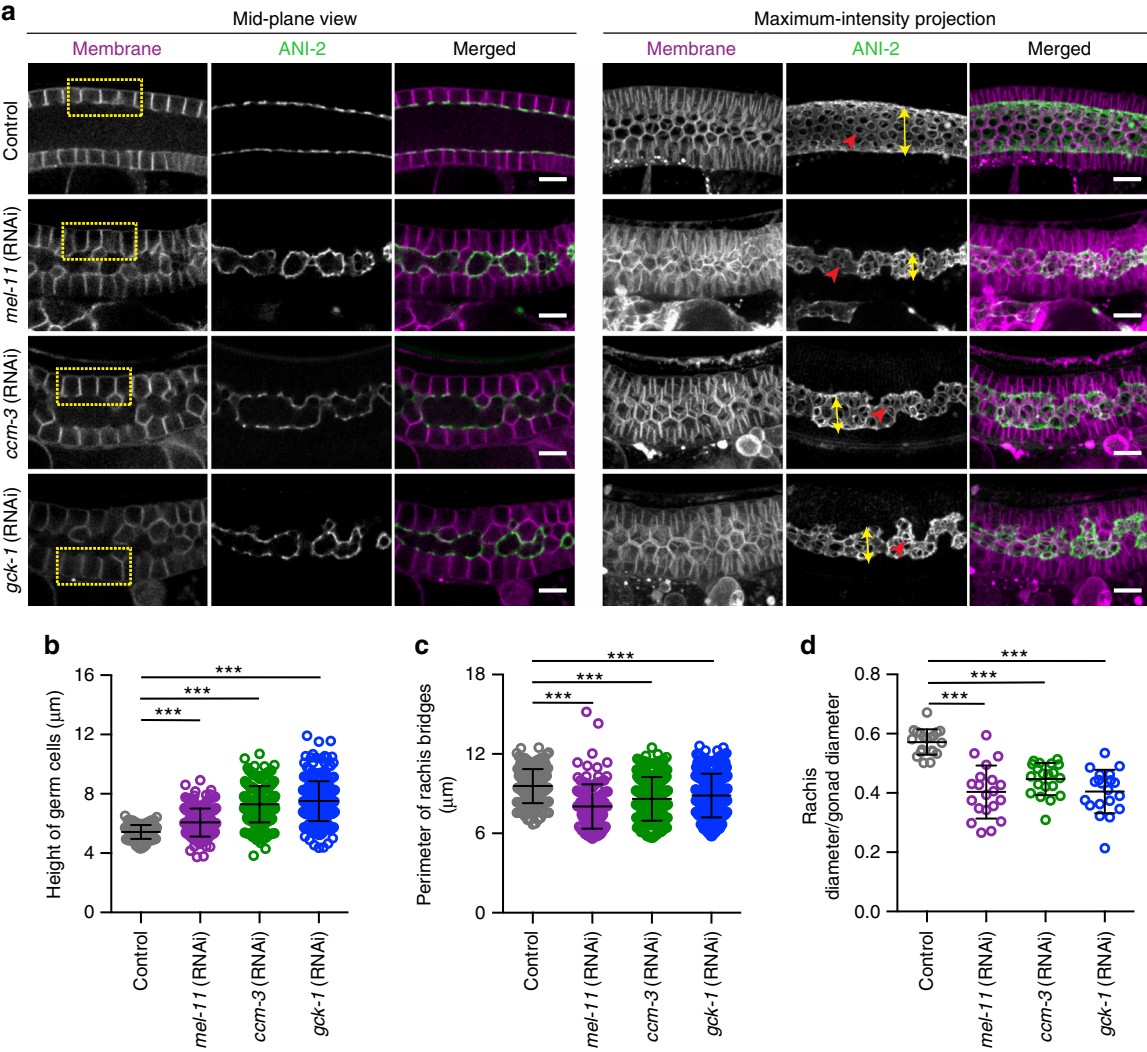

**Fig. 4** Excessive contractility results in taller germ cells, and overly constricted intercellular bridges and rachis. **a** Mid-plane and maximum intensity projection views of control, mel-11(RNAi), ccm-3(RNAi), and gck-1(RNAi) gonads expressing GFP::ANI-2 (green) and a membrane marker (magenta). Rectangular boxes (yellow) emphasize the increase in the height of germ cell membrane after knockdown. Red arrowheads highlight the difference in the perimeter of rachis bridges. Also the diameter of the rachis was reduced significantly in mel-11(RNAi), ccm-3(RNAi), and gck-1(RNAi) gonads (yellow double-headed arrow). **b** Quantification of the height of germ cells in the early meiotic region of the gonads of control ($n = 102$ membranes, $N = 10$ gonads), mel-11(RNAi) ($n = 196$ membranes, $N = 20$), ccm-3(RNAi) ($n = 250$ membranes, $N = 25$), and gck-1(RNAi) ($n = 250$ membrane, $N = 25$). **c** Quantification of the perimeter of rachis bridges in the early meiotic region of the gonads of control ($n = 187$ bridges, $N = 20$), mel-11(RNAi) ($n = 125$ bridges, $N = 16$), ccm-3 (RNAi) ($n = 307$ bridges, $N = 23$), and gck-1(RNAi) ($n = 307$ bridges, $N = 21$). **d** Quantification of rachis/gonad diameter in the early meiotic region of the gonads of control ($N = 20$), mel-11(RNAi) ($N = 20$), ccm-3(RNAi) ($N = 20$), and gck-1(RNAi) ($N = 20$). Error bars denotes s.d. Data represent mean ± s.d. of three independent experiments. Statistical analysis was done using one-way ANOVA test (***$p < 0.0001$). $N =$ number of gonads analyzed. Scale bar, 10 μm

on the presence of laminin, a basement membrane component, surrounding the entire germline indicating that the germ cells adhere strongly to a stiff basement membrane at their basal sides (Supplementary Fig. 4c). Periodic boundary conditions were applied along the tube's longitudinal extremities (Supplementary Notes 1 and 2).

With this model, we first computed the equilibrium shape of the wild-type tube (Fig. 7b). In this case, apical contractility was determined by the balance of radial-force contributions from the lateral sides and the apical actomyosin ring, which sets the wild-type apical contractility $\alpha_{WT} \approx 0.242$. Next, reducing the relative apical contractility ($\alpha/\alpha_{WT} < 1$) resulted in a tubular morphology with decreased cell height, wider rachis, and increased apical cell perimeter (Figs. 7c, d, 3d–g, Supplementary Movie 9). Increasing

the relative apical contractility ($\alpha/\alpha_{WT} > 1$) had the opposite effect, i.e. increased cell height, narrower rachis bridges and rachis, as observed in the experiments (Figs. 7c, e, 4a–d, Supplementary Movie 10). We next systematically varied both the relative apical contractility $\alpha/\alpha_{WT}$ and cell compressibility $\psi$. We thereby obtained a morphological phase-diagram ($\alpha/\alpha_{WT}$, $\psi$) which shows two branches of minimal-energy solutions demarcating two distinct phenotypic morphologies: (i) the wide-tube regime and (ii) the narrow-tube regime (Fig. 7f, Supplementary Fig. 4d–g, and Supplementary Note 3). Experimentally observed phenotypes are indicated in Fig. 7f by the isolines corresponding to experimental values of the average rachis diameter relative to gonad diameter (wild-type $d_r/d_g = 0.56$, cyk-1(ts) $d_r/d_g = 0.75$ and mel-11 (RNAi) $d_r/d_g = 0.41$) (grey dashed lines in Fig. 7f). The transition between

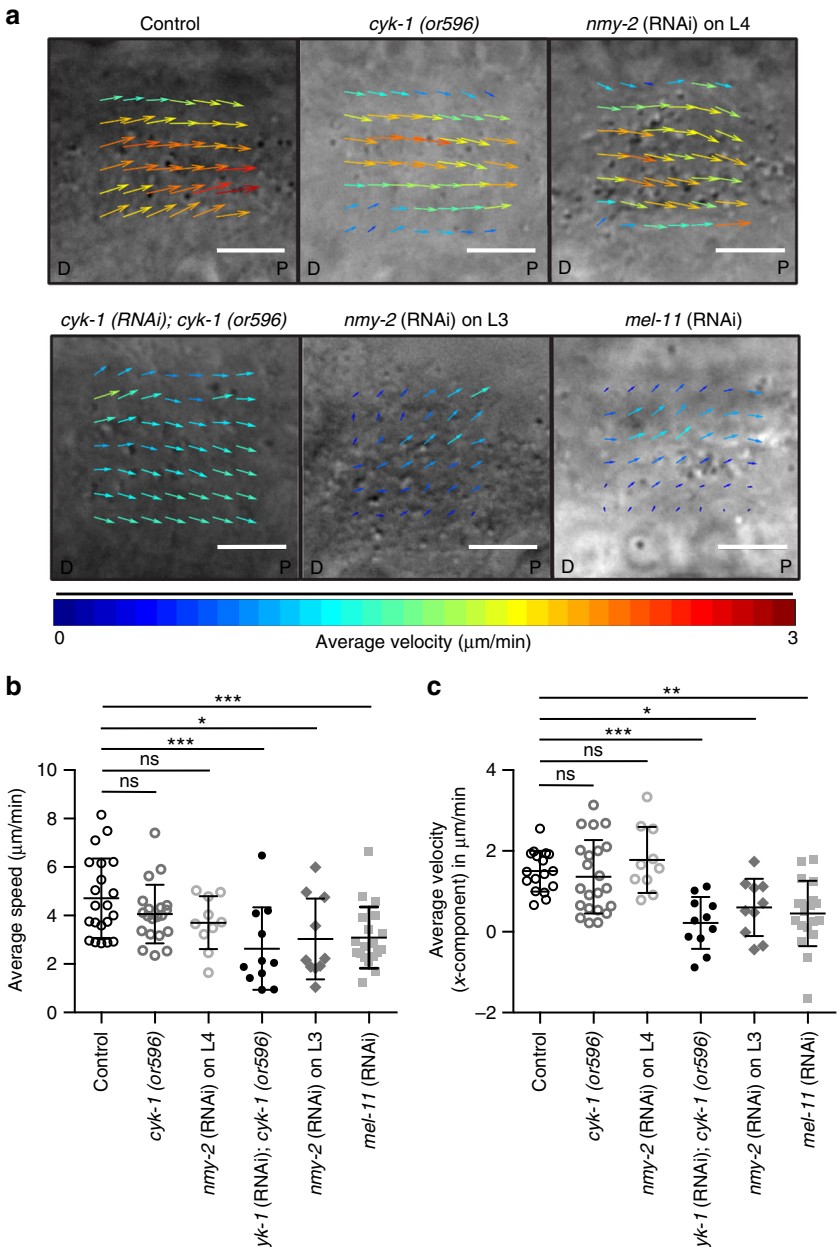

**Fig. 5** Loss of actomyosin regulators results in reduced cytoplasmic streaming. **a** Representative images of the average velocity vectors (PIV analysis) of the cytoplasmic streaming in the distal region of control, cyk-1(or596) at 20 °C, nmy-2(RNAi) on L4 stage worms, cyk-1(RNAi) on cyk-1(or596), nmy-2 (RNAi) on L3 stage worms, and mel-11(RNAi) gonads. Images are oriented in the direction of distal (D) to proximal axis (P) of the gonad. **b, c** Average speed and average velocity of the cytoplasmic streaming (along the X-axis) in the distal region of control ($N = 17$), cyk-1(or596) at 20 °C ($N = 22$), nmy-2 (RNAi) on L4 stage worms ($N = 10$), cyk-1(RNAi) on cyk-1(or596) ($N = 11$), nmy-2(RNAi) on L3 stage worms ($N = 10$), and mel-11(RNAi) gonads ($N = 19$). The particle speed and velocity was analyzed for 200 s in each movie. Data represents mean ± s.d. of four independent experiments. Statistical analysis was done using one-way ANOVA test (*$p < 0.05$, **$p < 0.005$, ***$p < 0.0001$, ns, non-significant), $N$ = number of gonads analyzed. Scale bar, 5 μm

the experimentally observed morphologies can be viewed as a trajectory between isolines in the graph ($\alpha/\alpha_{WT}$, $\psi$) (Fig. 7f).

Apico-basal tension differences have previously been proposed to be instrumental for buckled morphologies, e.g. in intestinal tissues[42,43]. Strikingly, our model suggests that apico-basal asymmetry alone cannot cause the rachis to buckle and develop pearling-like morphology under the relevant geometry and boundary conditions. This is due to the presence of external spatial constraints, e.g. adhesion of cells to the basement membrane (Supplementary Fig. 4c). The absence of such constraints (e.g. detachment of the gonadal tube from the basement membrane) severely affects the structural integrity of

the gonadal tube (Supplementary Note 1). However, when the apical contractility was modulated along the length of the tube, our model-tissue displayed a pearling-like morphology as observed for the gonads of plst-1(tm4255) with uneven distribution of contractility (Figs. 7g, 6f, Supplementary Fig. 5, and Supplementary Movies 11 and 12). It is worth noting that the small variations in the morphology of germ cells between the experiment and model might arise in part due to other contributions, such as disorder within the cell packing or local variations of the cortical activity. We tested the contribution of these factors in our model and the results show that these disparities did not have any significant effect on the overall shape

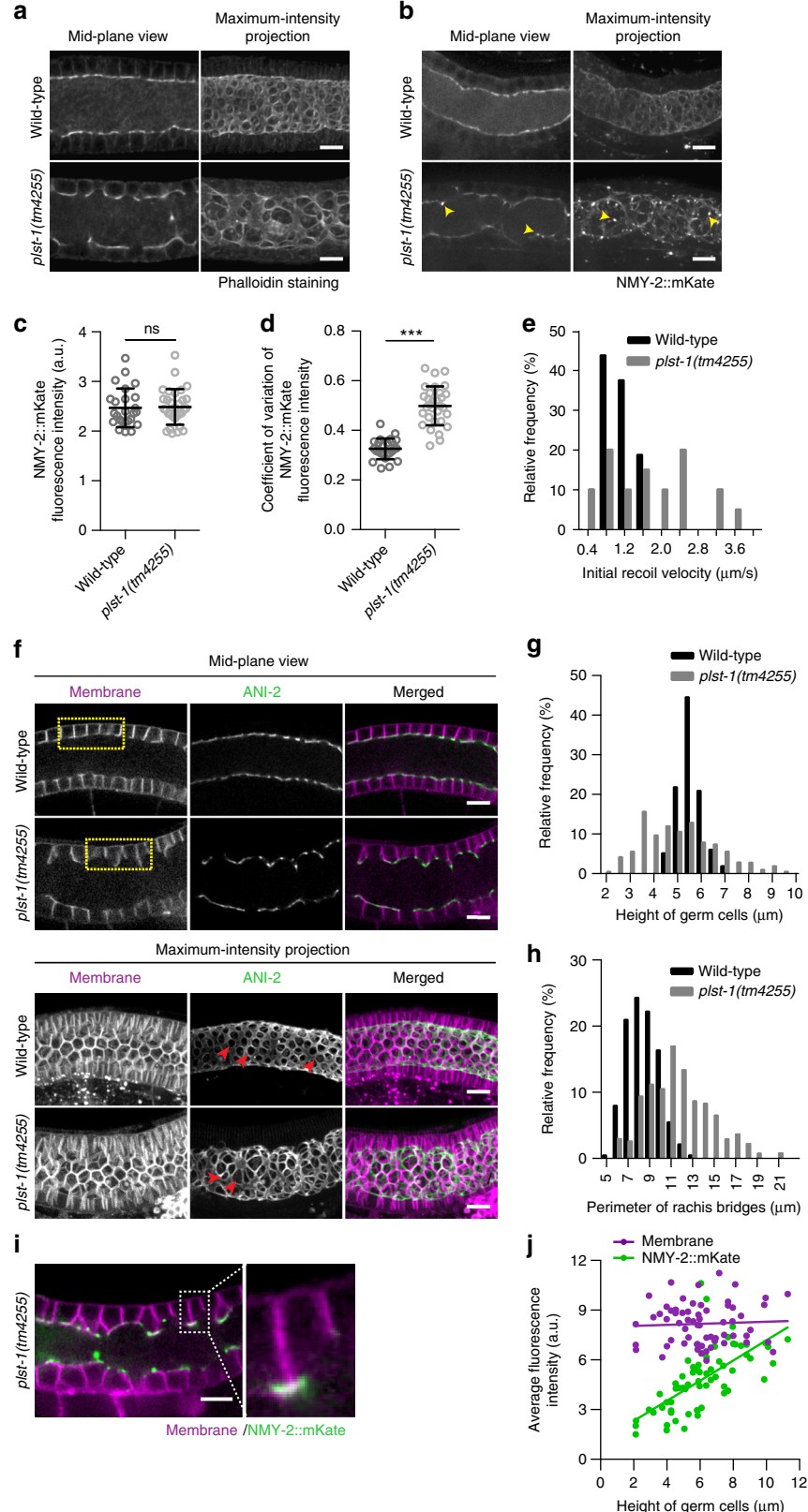

of the gonad under different conditions (Supplementary Fig. 6 and Supplementary Note 4). Together, these results further reinforce the significance of the actomyosin corset-like structure in the maintenance of germline architecture and regulation of global morphology simply based only on the variation of the apical contractility.

## Discussion

Although many genes required for the maintenance of germline structure have been reported[44], the mechanics of the germline remained unexplored. In the current study, we investigated the mechanics and function of a contractile actomyosin structure lining the rachis inside the *C. elegans* gonad. Alterations in

**Fig. 6** PLST-1, an actin-bindling protein, supports germline architecture by maintaining a uniform distribution of tension. **a** Representative mid-plane and maximum-intensity projected images of the phalloidin-stained gonads of wild-type and plst-1(tm4255) mutant worms. **b** Representative mid-plane and maximum-intensity projected images of the wild-type and plst-1(tm4255) mutant germline expressing NMY-2::mKate. Arrowheads (yellow) point to aggregates of myosin observed in plst-1(tm4255) mutants. **c** Average NMY-2::mKate intensity measured along the rachis in the early meiotic region of wild-type (N = 25) and plst-1(tm4255) mutant (N = 31) gonads. **d** Comparison of coefficient of variation (COV) of myosin intensity along the rachis measured in early meiotic region of wild-type (N = 25) and plst-1(tm4255) mutant (N = 31) gonads. Coefficient of variation is the ratio of standard deviation and mean fluorescence intensity. **e** Relative frequency distribution of initial recoil velocity of germ cell membrane after laser ablation at the rachis of the wild-type (N = 20) and plst-1(tm4255) mutant gonads (N = 20). **f** Mid-plane and maximum intensity projection views of the wild-type and plst-1(tm4255) mutant gonads expressing GFP::ANI-2 (green) and a membrane marker (magenta). The height of germ cells and distance between them were highly variable in plst-1(tm4255) mutant gonads than the wild-type (yellow rectangular boxes). Red arrowhead indicates variation in the perimeter of rachis bridges between wild-type and plst-1(tm4255) mutant gonads. **g–h** Relative frequency distribution of the height of germ cells and perimeter of rachis bridge in the early meiotic region of wild-type (N = 21) and plst-1(tm4255) mutant (N = 22). **i** Representative mid-plane image of the plst-1(tm4255) mutant germline expressing NMY-2::mKate (green) and a membrane marker (magenta). Enlarged image in the inset display the myosin level in the smaller and larger germ cell membrane. **j** Correlation between height of the germ cells and fluorescence intensity of GFP::PH and NMY-2::mKate in plst-1(tm4255) mutant gonads. Green dots denote fluorescence intensity of NMY-2::mKate and magenta dots indicates fluorescence intensity of GFP::PH at individual membranes (n = 64 membranes, N = 14 gonads). Error bar denotes s.d., data represents mean ± s.d. (***p < 0.0001, ns-non-significant) of three independent experiments. Statistical analysis was done using Mann–Whitney U-test, N = number of gonads analyzed and Scale bar, 10 μm

germline contractility have catastrophic effects on the functionality of the germline. Complete loss of contractility results in failure of cellularization and hence no gametes. On the other hand, excessive contractility results in the formation of smaller and non-functional oocytes. Oocyte growth predominantly depends on the cytoplasmic flow from the distal rachis into the proximal germ cells[9,10]. The mutants with excessive contractility have a hyper-constricted rachis at the distal as well as loop region and narrower opening of the distal germ cells (Fig. 4a–d and Supplementary Fig. 3), which along with a slow rate of cytoplasmic streaming (Fig. 5a–c and Supplementary Movie 8) likely results in insufficient transfer of cytoplasm, leading to the formation of small-sized oocytes. Furthermore, we found that actomyosin contractility is required for cytoplasmic streaming. A previous study suggested that force for cytoplasmic streaming is generated by or near the oocytes themselves[31]. Therefore, the reduced flow rate of cytoplasmic streaming in the absence of actomyosin regulators could be possibly due to reduced cortical tension within the oocytes. Whether tension at the inner corset enclosing the rachis also contributes to cytoplasmic streaming needs further investigation. Laser microsurgery in different directions along with genetic manipulations leading to partial loss of contractility and the computational model helped us to differentiate two sets of counter-balancing forces: in the plane of the rachis and perpendicular to it (radial tension). In the plane of the rachis, actomyosin contractility within the rachis rings counteracts tension within the rachis surface to control the proper germ cell opening into the rachis, as was also shown in previous studies[16,18]. Radial to the plane of the rachis, actomyosin contractility of the rachis counteracts membrane tension of the germ cells (and possibly hydrostatic pressure) to maintain their proper height.

The syncytial *C. elegans* gonad experiences a continuous mechanical stress due to flow of cytoplasmic material from the distal germ cells into the proximal oocytes[9] and large scale back and forth movement of the entire germline caused by contraction of sheath cells during ovulation[45]. How does the germline resist these mechanical strains while maintaining its architecture? ANI-2 has been suggested to confer elasticity to the rachis to help it withstand mechanical stresses[5]. We propose that the actomyosin-enriched corset helps to dynamically counterbalance changing forces and provides resistance against the mechanical strains crucial to maintain germline architecture. Given the remarkable conservation of several features

of syncytial germline[1–3], we speculate that a similar mechanism for the maintenance of germline architecture holds true across different metazoans.

## Methods

**Worm strain maintenance**. All strains were grown and maintained at 20 °C using standard protocol[46] except the temperature-sensitive strain of cyk-1(or596) which was maintained at 15 °C. Zhirong Bao's lab (Memorial Sloan-Kettering Cancer Center, New York, NY) provided strain BU70 [zbIs2(pie-1::lifeACT::RFP)]. Julie Canman's Lab (Columbia University, New York, NY) provided strain JCC389 [cyk-1(or596)]. Erin Cram's lab (Northeastern University, Boston, MA) provided strain UN1608 nmy-1(xb5[nmy-1::mKate2]). Nicolas T Chartier provided strain UM208 unc-119(ed3) III; ltIs81 [Ppie-1::gfp-TEV-Stag::ani-2; unc-119 (+)]; ltIs44 [Ppie1::mCherry::PH(PLC1delta1); unc-119(+)] IV. Endogenously tagged transgenic strains—COP1481 unc-119(ed3) III; unc-59 (knu463 [unc-59::degron::mKate2 +loxP unc-119 (+) loxP]) I, COP1234 cyk-4 (knu286 [cyk-4::GFP+loxP]) III and COP937 unc-119(ed3) III; cyk-1 (knu84 [cyk-1::GFP+loxP unc-119 (+) loxP]) III were generated by Nemametrix (Eugene, Oregon, USA) using CRISPR-cas9 genome editing technology. RZB282 strain was generated by crossing BU70 with LP162; RZB268 by crossing JCC389 with UM208; RZB267 by crossing RZB88 with LP229; RZB270 by crossing RZB88 with UM208; RZB266 by crossing LP229 with OD95 and RZB319 by crossing UM208 with NK651. The strains with their genotypes, used in this study, are listed in Supplementary Table 1.

**Generation of CRISPR-cas9 knock-in transgenic worm**. For making fluorescently tagged knock-ins, an established CRISPR-cas9 approach was used[47]. Briefly, a suitable Cas9 target sequence was designed using CRISPR design tool at http://crispr.mit.edu[48]. The target sequence was inserted into the Cas9–sgRNA construct (pDD162, Addgene#47549) using NEB's Q5 Site-Directed Mutagenesis Kit. The forward primer used was—5′N20GTTTTAGAGCTAGAAATAGCAAGT-3′, where N20 is the 20 bp targeting sequence from the design tool and reverse primer-5′-CAAGACATCTCGCAATAGG-3′. Two sgRNA guides were used to increase the efficiency of transgenesis by 95%. Next, a repair template was constructed by inserting 500–700 bp homology arms to the FP-SEC vector (FP—fluorescent protein, SEC— self-excising cassette) using NEBuilder HiFi DNA assembly mix from NEB. Injection mix containing 10 ng/μl of repair template, 50 ng/μl of Cas9-sgRNA construct with the target sequence, and fluorescent co-injection markers (10 ng/μl pGH8 (Prab-3::mCherry neuronal co-injection marker; Addgene #19359); 5 ng/μl pCFJ104 (Pmyo-3::mCherry body wall muscle co-injection marker; Addgene #19328), and 2.5 ng/μl pCFJ90 (Pmyo-2::mCherry pharyngeal co-injection marker; Addgene#19327)) were injected into the gonads of young adult N2 strain or unc-119(ed3) null strain worms. Three sets of 10 animals were injected. Hygromycin-resistant roller worms were selected by using 250 μg/ml hygromycin containing agar plates and then, heat-shocked to remove the selectable marker. Transgenic lines were finally confirmed by PCR and checking for fluorescence. For COP1481 strain, degron::mKate2 fluorescent tag was added to the C-terminus of the unc-59 sequence and for COP1234 and COP934, GFP tag insertion was made at the C-terminus of the native gene.

**RNA interference**. All RNAi experiments were done using the feeding method[49]. RNAi clones were obtained from Ahringer RNAi library[50] except NMY-1 and GCK-1. Gene-specific sequence of nmy-1 and gck-1 was amplified from genomic DNA and cloned into L4440 vector using Gibson assembly protocol[51] (NEB,

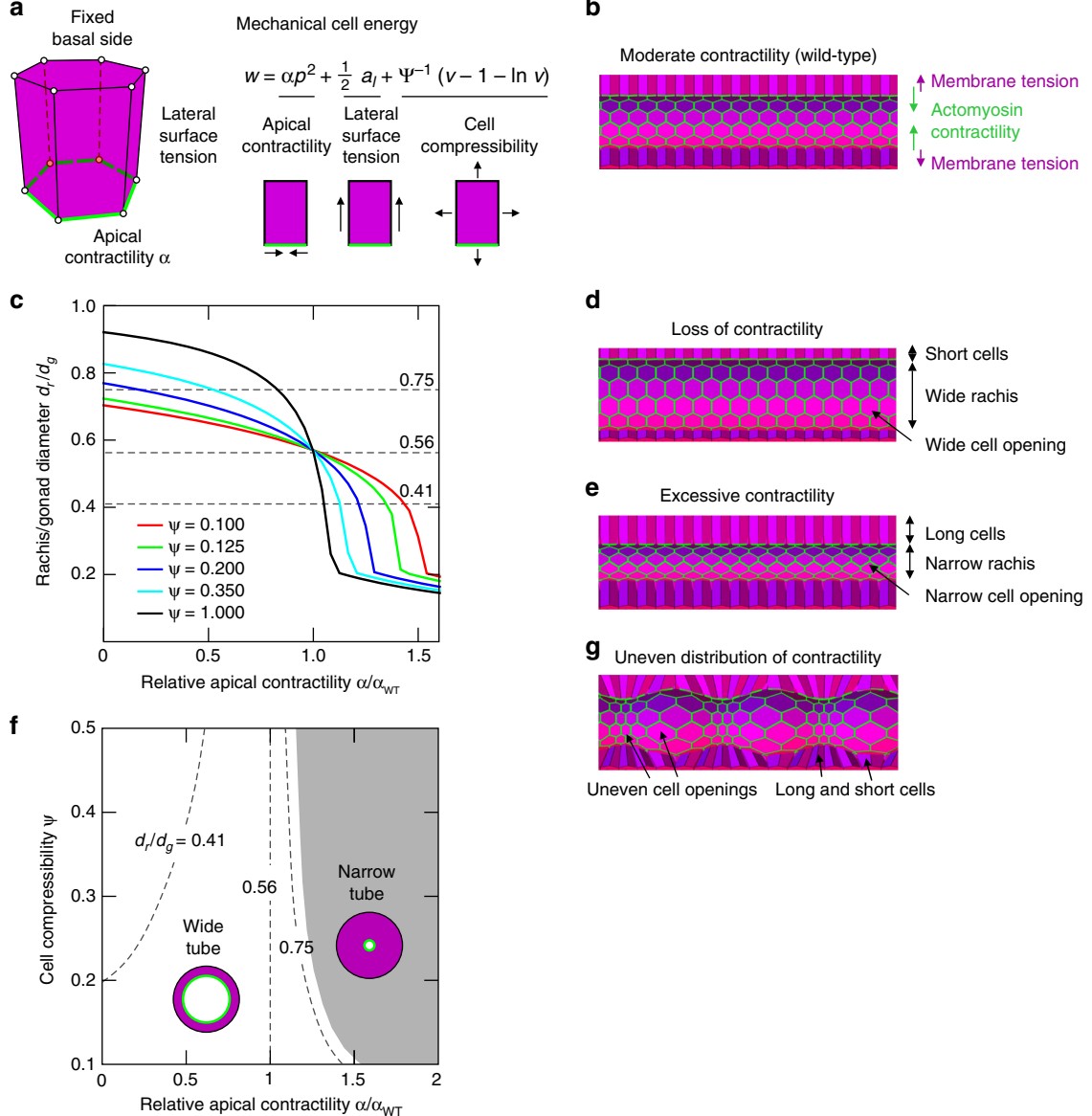

**Fig. 7** Mathematical model encapsulating the idea that syncytial germline architecture is regulated by contractile tension at the apical end of germ cells. **a** A three-dimensional computational model of the germline is based on 3D vertex model, where individual cells are modeled as polyhedral bodies with a polygonal base and apex. Forces were calculated from the prescribed potential energy $w$, which includes tension contributions from lateral cell surfaces, apical contractility, and cell compressibility. **b** Simulated germline tube morphologies at equilibrium for apical contractility $\alpha = 0.242$ at fixed compressibility, $\psi = 0.38$. **c** Rachis diameter relative to the gonad diameter $d_r/d_g$ at equilibrium as a function of the relative apical contractility $\alpha/\alpha_{WT}$ (where $\alpha_{WT}$ is apical contractility for wild type) is shown for varying values of cell compressibility $\psi$. The dashed lines at $d_r/d_g = 0.75$, 0.56, and 0.41 correspond for the three distinct phenotypical morphologies. **d** Simulated germline tube morphologies at equilibrium for apical contractility $\alpha = 0.121$ at fixed compressibility, $\psi = 0.38$. **e** Simulated germline tube morphologies at equilibrium for apical contractility $\alpha = 0.2701$ at fixed compressibility, $\psi = 0.38$. **f** Morphological phase diagram of the model tubular epithelium exhibits two distinct regimes: wide-tube and narrow-tube morphologies. Isolines connect morphologies that correspond to the experimentally measured average rachis diameters from *cyk-1*/formin mutant ($dr/dg = 0.75$), wild-type ($dr/dg = 0.56$) and *mel-11(RNAi)* ($dr/dg = 0.41$) worms. **g** Equilibrium tube morphology observed for an imposed contractile modulation along the apical boundary along the tube's axis for fixed compressibility, $\psi = 0.38$. The imposed modulation is $0.5\alpha_{WT} \sin(2\pi z/\lambda)$, where $z$ is the longitudinal coordinate of the tube and $\lambda$ is the tube's length

Catalog no. E2611S). Primers used for cloning are listed in Supplementary Table 2. RNAi plates were made using nematode growth medium containing 1 mM IPTG, 100 μg/ml of ampicillin, and 100 μg/ml carbenicillin. Overnight grown bacterial culture was diluted to 1:100 and grown at 37 °C until the culture reached log phase (~0.5OD). This culture was supplemented with 1 mM IPTG and grown at 37 °C for 3–4 h. After induction with IPTG, culture was seeded on the RNAi plates and appropriate stage worms were kept on the plates for depletion of mRNA transcript. For the partial knockdown of NMY-2, L4 stage worms were fed for 24 h while stronger depletion was achieved by feeding L3 stage worms for 48 h. Depletion of LET-502, MEL-11, CCM-3, and GCK-1 was done by feeding L3 stage worms for

48 h. For control, RNAi clone having L4440 vector without any insert was included in each experiment. All RNAi clones were confirmed by sequencing.

**Live imaging and analysis**. Intact worms were mounted with 5 μl of 10 mM levamisole on 3% agarose pad placed on a glass slide. For image acquisition, confocal microscope (Ti; Nikon, Tokyo, Japan) equipped with spinning-disk head (CSU-X1; Yokogawa, Tokyo, Japan) and Evolve or Prime95b camera (Photometrics, Tucson, AZ) was used. Images were acquired using Metamorph software (Molecular devices, Sunnyvale, CA). 40X or 60X 1.4 NA oil-immersion Plan-

Apochromat objective (Nikon, Japan) was used with Z-stacks (0.5 or 1 μm spacing) spanning through the entire thickness of the gonad.

The germ cell height was measured manually using Fiji[52]. Several germ cells were selected from the early meiotic region of the gonad (~80–100 μm from the distal end[16]) for all the measurements. Briefly, rachis and gonad diameter were measured by calculating average diameter from three distinct regions within the early meiotic region (~60–80 μm section). Perimeter of the intercellular bridges was measured from the projected images of a few consecutive Z-slices using an in-house Fiji macro as follows. First, for each Z-stack, a few consecutive Z-slices were maximum projected. Next, the image was smoothed by Gaussian blurring with sigma = 1, followed by Niblack local thresholding with radius of 35 pixels. The perimeter was then measured using Analyze Particles with size filter of 50–900 pixels and circularity filter of 0.45–1.00.

**FRAP analysis**. We used iLas2 targeted laser illumination system (Roper Scientific, Trenton, NJ) for the photobleaching experiments. A rectangular area at the rachis bridge was photobleached using 491 nm laser at 85% power with an exposure of 440 ms. Five Z-slices (1 μm spacing) were acquired using 100X 1.4 NA oil-immersion Plan-Apochromat objective (Nikon, Japan) at an interval of 30 s for the total duration of 8 min. Images for GFP::ANI-2 and PLST-1::GFP were acquired using 45% 491 nm laser power with 300 ms exposure time while NMY-2::mKate images were acquired using 35% 561 nm laser power with 200 ms exposure time. The average intensity for bleached region of interest (ROI) ($I_{frap}(t)$), background ROI ($I_{back}(t)$) and reference ROI ($I_{ref}(t)$) were measured using Fiji[52]. The intensity for bleached ROI was normalized using double normalization method[53] to correct for photobleaching and background. The normalized intensity, ($I_{norm}(t)$) was calculated as $I_{norm}(t) = \frac{I_{ref\_pre}}{I_{ref}(t) - I_{back}(t)} \cdot \frac{I_{frap}(t) - I_{back}(t)}{I_{frap\_pre}}$, where the subscript _pre indicates the average value before bleaching. Full scale calibration was applied to rescale the normalized values with the equation, $I_{norm\_sc}(t) = \frac{I_{norm}(t) - I_{norm}(t_{bleach})}{I_{norm\_pre} - I_{norm}(t_{bleach})}$, where $t_{bleach}$ is the bleach time. The rescaled values, $I_{norm\_sc}(t)$ were fitted with single exponential function, $I(t) = P(1 - e^{-kt})$, where $P$ is the mobile fraction and $k$ is the rate constant[54]. The half-life was computed as $t_{1/2} = -\ln(0.5)/k$ and the immobile fraction was computed as $1-P$ GraphPad Prism 6.0 (GraphPad Software, La Jolla, CA, USA) was used for curve fitting.

**Laser incision experiment and analysis**. The laser ablation system used in this study has been reported previously[56]. Worms were immobilized with 10 mM levamisole and mounted on 3% agarose on a glass slide. Line laser incision (~9 μm) using ultraviolet pulses of 90 nW (duration for line ablation was 3–4 s) was carried out on the rachis surface (~5–6 μm from the surface view of the gonad) visualized using a transgenic line co-expressing a rachis marker GFP::AN1–2 and membrane marker mCherry:: PH. For point ablation, ultraviolet pulse of 135 nW was used at the rachis bridge of individual cells (mid-plane view). Images were acquired at the rate of 1 frame per second for the duration of 5 s before the ablation and 1 min after the ablation using Nikon A1R MP confocal microscope attached to the laser ablation system.

Movement of the gonad within the worm was corrected by using drift correction function of the Imaris 8.4 software. Next, we tracked manually the distance between two nearby membrane edges at each time point, using MtrackJ plugin of the Fiji[52]. Before the laser ablation, the distance between two edges was fitted using a linear function. After ablation, we used a double exponential function to fit the data. We used a single exponential function in cases where the recoil velocity was small and a double exponential function overestimated the velocity. The effective time of laser cut was estimated as the intersection of linear and exponential fitting curves. The recoil velocity was calculated, using Matlab software, as a derivative of the exponential function at the effective time of laser cut[55,56].

Total displacement was calculated by measuring the difference between the initial distance between the lateral germ cell membrane and the final distance between them after 50 s of point ablation at the rachis bridge.

To track the circularity changes for each rachis bridge, an in-house MATLAB code was developed for segmenting, tracking, and measuring circularity. Segmentation was performed by Niblack local thresholding. Each segmented germ cell opening was then tracked across time by using the criteria of maximum overlapping. The circularity was calculated as $\frac{4\pi.area}{perimeter^2}$, where a value of 1 indicates a perfect circle and a decreased value indicates elongation.

**Fluorescent intensity measurement**. Mid-focal plane of the gonads were selected for the measurement of intensity of phalloidin, NMY-2::mKate, and GFP::PH along the rachis bridge. Before quantifying the intensities, each image was corrected for the background fluorescence. After background subtraction, intensity was measured by drawing a line (with 7-pixel width) along the rachis bridge using Fiji software[52]. Mean intensities of different samples of the control and test were compared.

**Colocalization analysis**. Analysis of colocalization of the actin (LifeAct::RFP) and non-muscle myosin-II (NMY-2::GFP) at the rachis bridge was performed using

ImarisColoc Module of Imaris 8.4 (Bitplane AG, Zurich, Switzerland). Before analysis, we used Fiji to select a region of interest (rachis bridge) and created a masked GFP channel using this ROI. Pearson's coefficient of correlation between NMY-2::GFP and LifeAct::RFP within this ROI was calculated using Imaris8.4.

**Phalloidin staining**. Phalloidin staining of the gonads was done using Method I reported earlier by Wolke et al.[9]. Briefly, worms were dissected in culture buffer (73 mM HEPES pH 6.9, 1.6% sucrose, 10 mM EGTA, 2 mM MgCl2, 40 mM NaCl, and 5 mM KCl) included with 0.5 mM levamisole on a glass slide. Extruded gonads were fixed in 3% formaldehyde prepared in culture buffer for 10 min. The fixed gonads were washed with PBS and then permeabilized in PBS containing 0.025% Triton-X100 for 5 min. Next, gonads were stained with 66 nM Alexa Fluor 488-conjugated phalloidin (Life Technologies) for 20 min at room temperature in dark conditions. After rinsing the phalloidin-stained gonads with PBS, they were mounted on the glass slide with vectashield mounting medium (Vector lab, Cat. no. H-1300) for fluorescence imaging.

**Pharmacological treatments**. Worms were microinjected with 5 μM Latrunculin A (Abcam ab144290), an inhibitor of actin polymerization, or 250 μM myosin light chain kinase (ML-7) inhibitor (Sigma, Cat. no. I2764) along with a DNA marker Hoechst dye (Hoechst 33342, Life Technologies H3570). Injection was done at the loop region of the gonad and distal region was imaged immediately after Latrunculin A injection, while for ML-7 treatments, gonads were imaged ~40 min after injection. Dimethyl sulfoxide (DMSO) was used as control for injection experiments.

**Cytoplasmic streaming**. To measure particle velocity in the early meiotic region of the gonad, intact worms were immobilized with 2–3 μl of 0.5 μm polystyrene beads (Polysciences, Cat. no. 08691) on 10% agarose pad placed on a glass slide. Differential interference contrast (DIC) movies were made at an interval of 2 s for 8–10 min using wide field microscope. The relative velocity of the particles with respect to the gonad movement was calculated as $\mathbf{v}_r = \mathbf{v}_p - \mathbf{v}_g$, where $\mathbf{v}_p$ is the particle velocity and $\mathbf{v}_g$ is the velocity of gonad movement. To measure $\mathbf{v}_g$, the gonad movement within the intact worms was tracked using the Track Objects Module of MetaMorph Offline 7.8.13 (Molecular devices, Sunnyvale, CA). As for $\mathbf{v}_p$, particle image velocimetry (PIV) analysis was performed using a MATLAB PIV toolbox, MatPIV 1.6.1 (http://www.mn.uio.no/math/english/people/aca/jks/matpiv/). Single-pass PIV with window size of 32 × 32 pixels and 50% overlapping windows was applied on a region of interest (10.75 μm × 10.75 μm) in the early meiotic region of the gonad of intact worms. Each relative velocity, $\mathbf{v}_r$ was resolved into component along the flow, $\mathbf{v}_{rx}$ and component perpendicular to the flow, $\mathbf{v}_{ry}$. The velocity component $\mathbf{v}_{rx}$ that is pointing from the distal end to the proximal end is defined as positive. The relative speed was computed as $s_r = \sqrt{\mathbf{v}_{rx}^2 + \mathbf{v}_{ry}^2}$. For each time point, $\overline{\mathbf{v}_{rx}}$ and $\overline{s_r}$ for the region of interest was computed and these values were averaged across time for each movie.

**Statistical analysis**. GraphPad Prism 6.0 was used for performing the statistical tests. We used Mann–Whitney U-tests (for comparing two samples) or one-way analysis of variance (ANOVA; for comparing more than two samples) to calculate p values. p-Value < 0.05 was considered as significant. All the graphs are represented as mean ± s.d. otherwise indicated. The exact numbers of samples are indicated in the figure legend of each experiment. All the experiments were repeated at least 3–4 times and each data set shown is the representative data.

**Code availability**. All codes written in Matlab for analysis of cytoplasmic streaming experiment, PIV analysis, changes in circularity and measuring recoil velocity are available upon request.

## Data availability
All the relevant data and reagents are available upon request from the corresponding author.

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

## Acknowledgements

This work was supported by Singapore Ministry of Education Tier 2 grant MOE2015-T2-1-045 awarded to R.Z.B. and MOE2015-T2-1-116 awarded to Y.T. We thank Erin Cram and Alison Wirshing for sharing the mKate::NMY-1 strain. We thank Zhirong Bao for providing BU70 strain [*zbIs2(pie-1::lifeACT::RFP)*] and Julie Canman for sharing JCC389 strain [*cyk-1(or596)*]. We thank Yusuke Hara for help with laser ablation experiments. We thank Amy Maddox for helpful discussions. We thank MBI communication core for help with illustrations. Some strains were provided by the CGC, which is funded by NIH Office of Research Infrastructure Programs (P40 OD010440).

## Author contributions

R.Z.-B. conceived the project. P.A. and R.Z.-B. designed experiments, analyzed results, and wrote the manuscript. P.A. performed all experiments. H.T.O. contributed image analysis tools. Y.T. assisted with laser ablation experiments and analysis. A.P. assisted with pharmacological perturbation experiment. S.D. and M.K. designed the theoretical

model, performed in silico numerical simulations and wrote modeling sections of the manuscript.

## Additional information

**Competing interests:** The authors declare no competing interests.

