## [Peer Review File · Nature Communications]

Reviewers' comments:

Reviewer #1 (Remarks to the Author):

The manuscript by Priti et al seeks to characterize the syncytial organization of the adult *C. elegans* germline. The authors first validate previous observations regarding the localization of various contractility regulators on the surface of the rachis. FRAP measurements of 3 of these regulators reveals that their turnover is slower than in cytokinetic rings and variable. Measuring recoil after laser cutting reveals that the rachis rings are under myosin-dependent tension, with higher tension in the distal end. Depletion of various positive or negative contractility regulators or the use of drugs perturbing actin or myosin affects germ cell morphology and rachis ring diameter. Perturbing these regulators also perturbs cytoplasmic flows that drive oocyte expansion. Finally they present a computational model of the rachis that recapitulates some of their observations and propose that actomyosin-dependent force imbalance in the germline drive oocyte expansion by causing changes in hydrostatic pressure within the tissue.

Overall the work is original and well done, and the manuscript contains many results that describe an interesting phenomenon that should appeal to cell and developmental biologists in general. However much of the work appears as largely correlative and, in some cases, results are overinterpreted or lack mechanistic depth, which limits my enthusiasm.

1. One of my main concerns is regarding the conclusion that there are both axial (at rachis bridges) and radial (orthogonal) forces acting on the rachis to maintain its organization. This conclusion rests largely on the assumption that there is a direct correlation between surface tension (as measured by rachis deformation following laser cutting) and the changes in cell shape that are documented in the manuscript. While the laser ablation experiments that are performed in Figures 1 and S1 clearly reveal that the system is under tension, it does not allow to determine whether tension is due to axial or radial forces. The relative contribution of these forces is inferred by measuring changes in cell shape that occur following depletion of various actomyosin contractility regulators and correlating this with the localization of these markers, including in the experiments where the localization and activity of these regulators is more random (in *plst-1* mutants). Yet all of this rests on correlative evidence that changes in cell shape are imparted by changes in tension. This assumption is somewhat substantiated by the *plst-1* mutant analysis but this constitutes a single condition and therefore it is unclear if this relationship can be generalized. One way to address this could be by varying the method for laser cutting. For instance, my understanding is that all cuttings were done on rachis bridges, either by point ablation or by line ablation in the distal-proximal axis. Were there cases where cuts were done without sectioning a rachis bridge (i.e. on a region of the rachis that does not have bridges)? This would be informative to assess radial tension when compared to similar cuts done across rachis bridges and point ablations. Likewise, cutting across the distal-proximal axis of the germline would be a good control to have, as it should have minimal lateral recoil (and perhaps a pronounced longitudinal recoil) based on the authors' conclusions. Perhaps the model could be used to formulate a hypothesis in this regard (see below).

2. The link between rachis tension and the cytoplasmic streaming that is caused by a change in hydrostatic pressure is also tenuous. The authors make this link by perturbing contractility regulators and correlating changes in cell shape with changes in cytoplasmic streaming, yet the experiments are too crude to make such a connection here. One way to block streaming that was previously used by the Wolfe/Priess study was to inject a drop of oil in the rachis. This experiment revealed that flows were still active proximal to the injection site but not distal. Perhaps this simple condition to block the cytoplasmic streaming and decouple distal and proximal streaming could be used to further monitor tension within the rachis (using laser cutting) and better link changes in rachis tension with hydrostatic pressure. Without this I am not sure what is the significance of the results reported in Figure S3. Also, the finding by Wolfe et al. that the force generating streaming is in the proximal region is difficult to reconcile with the authors' finding that there is more tension in the distal region compared to the proximal region. This should be addressed.

3. I do not understand the significance of the FRAP analyses performed in Figure S1. While the

experiments are well done, what is the significance of different actomyosin contractility components having different turnover rates at rachis bridges? How does this impact force generation within the rachis? Perhaps the authors could incorporate this information in their in silico model and use it to make predictions that can be experimentally tested? As written I do not understand what this brings to the story and/or what I should conclude from these experiments other than the results stated.

4. I do not have proper mathematical or computational expertise to assess whether or not the in silico model is correct. However, while I appreciate that the model recapitulates some of the observations that were made, I find it strange that the model is not tested per se. In my opinion, a model proves correct when it makes a prediction that can be experimentally verified, which is not the case here. In this sense I think that the model could be better exploited, for instance using the suggestions above. Also, in silico modeling of forces in the rachis was previously carried out (Coffman et al., *Biophys. J.* 2016). While the authors document discrepancies between their work and that published in this article (with NMY-1 localization and phenotype), which suggests that this other model may rest on false premises, I think that it would be important to comment on the comparison between their approach to model the syncytial germline and this one.

Minor comments are suggestions:

5. Line 44: The authors state that *C. elegans* has 2 U-shaped gonads. The convention is that *C. elegans* has a single gonad that forms 2 U-shaped gonad arms.

6. In the first paragraph of p. 5, the authors conclude that "more than half of the tension in the germline is myosin-dependent" based on laser-cutting experiments done in animals depleted of *nmy-2*. As stated, this conclusion can be misleading because there is an underlying assumption that some of the tension is myosin-independent, which is clearly not possible to conclude from these results. I suggest rewording this section to make sure that it is clear that *nmy-2* is partially depleted and therefore this only highlights the fact that myosin is a significant contributor to the tension (it could even account for all of it).

7. The authors refer to the "contractome" when discussing contractility proteins. I am not quite sure what is encompassed by the term "contractome". All previous work have referred to contractility regulators or proteins when discussing about the proteins studied here. I would therefore keep nomenclature straight and use "contractility regulators", so as to avoid building confusion within the field.

8. The genotype of many of the strains listed in Table S1 is not correct, namely COP1481, COP1234, COP937 and UN1608. Please use proper nomenclature.

Reviewer #2 (Remarks to the Author):

What is the mechanical basis for the maintenance of syncytial architecture, an important and conserved aspect of the germline? The present study tackles this question by focusing on the role of actomyosin contractility in the maintenance of germline syncytial organization in *C. elegans*. The authors employ an interdisciplinary approach, combining live fluorescence confocal microscopy of the early meiotic region of a gonad under genetic, pharmacological and mechanical perturbations with computational cell-based modelling of the common rachis. Based on their results, the authors conclude that in the plane of the rachis, actomyosin contractility counteracts tension to control the correct germ cell opening into the rachis, while in an orthogonal direction, contractility counteracts hydrostatic pressure and membrane tension to maintain the correct germ cell heights. Overall, this work demonstrates the role of actomyosin contractility in maintaining syncytial architecture.

As a modeller, I am not well placed to comment on the validity of the experimental techniques used, nor their biological significance. However, I found this manuscript to be well written, the study itself to be clearly motivated, and the findings interesting. While no quantitative predictions were generated by the computational model, this nevertheless represents an innovative and complementary approach to studying the mechanics underlying syncytial architecture. In my view,

this work would be of interest to researchers working in other areas of epithelial mechanics and morphogenesis.

I have the following technical queries on the main text:

p.9 "We propose the flow could also be driven by a gradient of hydrostatic pressure generated by the higher rachis contractility at the distal region compared to proximal" - Do the authors' simulations bear out this hypothesis? Presumably they could visualize the pressure in each cell in a relevant simulation (related to the fourth contribution to the author's choice of energy functional in equation (S3) in the supplementary information)?

p.11 "The gonadal tube was modeled as a cylindrical epithelial sheet of closely packed cells with hexagonal bases and apices" - Is this geometric assumption reasonable, based on the authors' live imaging data? From eyeballing Figure 1 it seems that the majority of cells are indeed hexagonal, though presumably it would be straightforward to quantify this. In either case, an additional note clarifying whether this is based on the experimental data, or an assumption designed to simplify the model calculations, would be helpful (would the authors expect their simulations results to be affected at all if cells were not all hexagonal?).

In addition, I have the following queries on the supplementary information for their 3D vertex model:

p.2: "We assume a first-order dynamics given by the overdamped equation of motion for vertices" - As noted by the authors just after equation (S5), the assumption is made that all vertices have the same friction drag coefficient. Since this term is proportional to the velocity of each vertex, it would seem to represent friction against the underlying matrix. Is there extracellular matrix or a basement membrane either apical or basal to the germ cells? If so, is this uniform in space; if not, could a cell-cell friction term be more suitable?

p.4 "The basal side of the gonadal tube is fixed" - Is this assumption based on the authors' experimental observations? Given their findings concerning the lack of buckling/undulations in simulations under differences in apico-basal tension, I am wondering if this behaviour might occur if the basal surface were allowed to deform to some extent?

Finally, I spotted the following minor typographical errors:

p.8: "in two orthogonal direction" -> "in two orthogonal directions"

p.17: "The germ cells enter mitosis at the distal end and progresses" -> "The germ cells enter mitosis at the distal end and progress"

p.17: "to form mature oocyte" -> "to form a mature oocyte" or "to form mature oocytes"

Methods: "Height of the germ cells" -> "The germ cell heights"

Supplementary Video Legends: "Green hexagons represents apical end of the germ cells" -> "Green hexagons represent germ cells' apical ends"

Reviewer #3 (Remarks to the Author):

Priti et al 2018 submitted to Nature Communications

This manuscript focuses on the role of actomyosin contractility in maintaining the organization of the syncytial gonad of *C. elegans*. The authors do a very nice job in describing the structure of a corset-like sheath composed of Actin, Myosin and several of their regulators that separates forming oocytes from a central, tubular shaft or rachis. Rachis bridges form a direct connection between the oocytes and the rachis and are somewhat similar to the ring canals that connect germ cells in the *Drosophila* ovary. Components of an actomyosin "contractome" that co-localize with actin and myosin in the sheath include anillin, the actin bundling protein PLST-1, a septin, a

RhoGAP. The authors used FRAP analysis to evaluate protein stability in the corset and find that different components recover from photobleaching at different rates and have different characteristic mobile fractions. Nevertheless, all proteins tested exchanged more slowly than from the cortex of cells, suggesting that the rachis sheath is particularly stable, with slow turnover and large, immobile pools. In addition, the authors used laser ablations to show that the actomyosin corset is under tension, both circumferentially and longitudinally (however, see specific comments below). Moreover, they show that various strategies designed to reduce the level of actomyosin contractility (e.g., myosin knockdown or the use of a temperature sensitive formin allele) reduce sheath tension. Further analysis of reducing actomyosin contractility through depletion of NMY-2 (but not NMY-1) could, at extreme levels cause sterility, but at intermediate levels caused the height of the germ cells to decrease (because the sheath was less contractile) and caused the diameter of the rachis bridges to increase. Similar results were caused by compromising the activity of the Rho kinase, Let-502 or the formin, *cyk-1* and pharmacological strategies to compromise actomyosin function also perturbed corset structure. In contrast, depletion of negative regulators of actomyosin function had the opposite effect (see query in the specific comments below). The authors further show that cytoplasmic flow is reduced when actomyosin contractility is compromised and suggest that the flow is driven by a hydrostatic pressure gradient caused by increased rachis contractility in the distal region of the gonad. A final series of experiments focused on the morphology of gonads from worms defective in *PLST-1*, a plastin homolog that assures a smooth distribution of actin and myosin, and actomyosin based tension in the rachis and rachis bridges. Finally, the authors generate an in-silico model for germline architecture based on a 3D vertex approach and show the model recapitulates the steady state shape of the gonadal tube. Moreover, increasing or decreasing in silico tension recapitulates changes in rachis diameter, consistent with experiment. Introduction of a tube with apical tensions that vary periodically further mimic the effects of *PLST-1* mutants.

Overall, the experiments and the model confirm the importance of the contractile actomyosin sheath in maintaining gonad structure. Ultimately, the paper should make a fine contribution to Nature Communications. Nevertheless, the specific comments below should be addressed before it can be published.

Specific comments.

Lines 94-105. The authors report an initial recoil velocity for a variety of conditions, but do not report the time-course of recoil. Was distance of recoil vs. time linear? Exponential? A sum of two exponentials? In the supplement, the authors state that "Initial recoil velocity was obtained by calculating the average recoil velocity in the first ten seconds after the ablation." This only makes sense if the recoil was linear with time and averaging was designed to account for noise. It might possibly be a very poor measure of initial recoil velocity if recoil decay is exponential or the sum of exponentials. The authors need to tell the reader what the recoil looked like (present distance vs. time in a supplemental figure) and convince the reader that their method of calculating an average is appropriate.

The 8 higher magnification panels in Figure 1g do not have an associated scale bar and one should be added.

Line 107. The authors state "Next, we sought to decrease the level of actomyosin contractility and observe the effects on syncytial germline structure." This statement is somewhat confusing given that line 96 deals with the depletion of myosin activity and lines 99-103 deal with the effect on recoil of natural changes in the level of myosin phosphorylation and therefore activity.

Line 112. Why do the authors introduce *cyk-1* here, but wait until line 490 to tell us it's a formin?

Line 185. The authors need to explain why "excessive contractility", which, according to line 168 causes a small but significant increase in the height of the germ cells would end up causing small, non functional oocytes. Shouldn't a smaller rachis and taller germ cells give rise to taller oocytes?

Perhaps the authors need to explain better what happens in the proximal region as the oocytes transition through the U turn that the gonad makes? This comes up again on line 291-292.

Finally, the in silico model, which is detailed in the Supplement is likely to be attractive to many mathematicians interested in morphogenesis. It could be presented in a much more useful fashion and should be. Specifically, the model appears to fit well the biology being studied. Nevertheless, like all models it requires simplifications. The authors are best suited to explain which simplifications fit the biology almost precisely, and which fall short and how. Further, the model as written (like many such models) seems to be written for experts in modeling. The description of the model in the main text is fine. Both additions to the model, a more biologically friendly description of the model and a careful analysis of how each of the assumptions that allow simplification maps to the biology could come simply by interspersing each section devoted to the model in the supplementary material with appropriate expansions of the supplementary text, and where useful, additional supplementary figures.

Response to reviewers' comments:

Reviewer #1 (Remarks to the Author):

The manuscript by Priti et al seeks to characterize the syncytial organization of the adult C. elegans germline. The authors first validate previous observations regarding the localization of various contractility regulators on the surface of the rachis. FRAP measurements of 3 of these regulators reveals that their turnover is slower than in cytokinetic rings and variable. Measuring recoil after laser cutting reveals that the rachis rings are under myosin-dependent tension, with higher tension in the distal end. Depletion of various positive or negative contractility regulators or the use of drugs perturbing actin or myosin affects germ cell morphology and rachis ring diameter. Perturbing these regulators also perturbs cytoplasmic flows that drive oocyte expansion. Finally they present a computational model of the rachis that recapitulates some of their observations and propose that actomyosin-dependent force imbalance in the germline drive oocyte expansion by causing changes in hydrostatic pressure within the tissue.

Overall the work is original and well done, and the manuscript contains many results that describe an interesting phenomenon that should appeal to cell and developmental biologists in general. However much of the work appears as largely correlative and, in some cases, results are overinterpreted or lack mechanistic depth, which limits my enthusiasm.

1. *One of my main concerns is regarding the conclusion that there are both axial (at rachis bridges) and radial (orthogonal) forces acting on the rachis to maintain its organization. This conclusion rests largely on the assumption that there is a direct correlation between surface tension (as measured by rachis deformation following laser cutting) and the changes in cell shape that are documented in the manuscript. While the laser ablation experiments that are performed in Figures 1 and S1 clearly reveal that the system is under tension, it does not allow to determine whether tension is due to axial or radial forces. The relative contribution of these forces is inferred by measuring changes in cell shape that occur following depletion of various actomyosin contractility regulators and correlating this with the localization of these markers, including in the experiments where the localization and activity of these regulators is more random (in *plst-1* mutants). Yet all of this rests on correlative evidence that changes in cell shape are imparted by changes in tension. This assumption is somewhat substantiated by the *plst-1* mutant analysis but this constitutes a single condition and therefore it is unclear if this relationship can be generalized. One way to address this could be by varying the method for laser cutting. For instance, my understanding is that all cuttings were done on rachis bridges, either by point ablation or by line ablation in the distal-proximal axis. Were there cases where cuts were done without sectioning a rachis bridge (i.e. on a region of the rachis that does not have bridges)? This would be informative to assess radial tension when compared to similar cuts done across rachis bridges and point ablations. Likewise, cutting across the distal-proximal axis of the germline would be a good control to have, as it should have minimal lateral recoil (and perhaps a pronounced longitudinal recoil) based on the authors'*

conclusions. Perhaps the model could be used to formulate a hypothesis in this regard (see below).

Response: We thank reviewer #1 for their appreciation of the novelty, high quality, and wide appeal of our work and for their points of criticism, which we have addressed in the revised manuscript, as detailed below.

Following the reviewer suggestion, we have substantially expanded our laser ablation experiments (new figure 2) to probe tension in all directions. To better characterize in-plane tension we now include point ablations in regions of the rachis between bridges, and line ablations in a direction perpendicular to the previously analyzed distal-proximal axis. Representative movies and images are shown in supplementary movie 5 and figure 2e,f, and quantification of displacement and initial recoil velocities calculated from 15 gonads for each ablation type is shown in figure 2h,i. The displacement and initial recoil velocity from these ablations was identical to ablations carried out along the direction of distal and proximal axis (Recoil velocity after line ablation along the rachis surface- $0.68 \pm 0.12 \mu\text{m}/\text{second}$, Point ablation- $0.69 \pm 0.12 \mu\text{m}/\text{second}$ and Perpendicular line ablation- $0.68 \pm 0.14 \mu\text{m}/\text{second}$.) demonstrations that tension in the plane of the rachis is uniform in all directions.

In order to probe the radial tension, as suggested by the reviewer, we did line ablations across the germ cell membranes holding the rachis surface (without sectioning the rachis bridge or rachis surface) (Supplementary movie 6 and figure 2g). Indeed, we found a rapid displacement of the nearby rachis surface and a gradual change in the shape of the rachis bridges with time. The shape of the rachis bridge present away from the region of laser ablation remains unaltered (Fig.2 g,j).

Taken together, these results along with the genetic perturbations show the presence of two forces that maintain germline architecture– one acts along the rachis surface to close the rachis bridges and another one acts radially to the rachis to maintain appropriate height of the germ cells.

In the model, mechanical balance of forces at every vertex determines cell shape at any time point and all forces are due to tensions prescribed. Furthermore, we have validated the boundary conditions and justified by additional numerics and experiments. Hence, from the perspective of the vertex model it is imposed that *“changes in cell shape are imparted by changes in tension”*. *In vivo*, direction of tension is inferred from the recoil obtained from laser ablation experiments and the changes in cell shape after genetic manipulations whereas, in the simulations we impose the tension to obtain the equilibrium cell shape as the output. Therefore, it is not possible to determine the recoil velocity under varying conditions from our model.

2. *The link between rachis tension and the cytoplasmic streaming that is caused by a change in hydrostatic pressure is also tenuous. The authors make this link by perturbing contractility regulators and correlating changes in cell shape with changes in cytoplasmic streaming, yet the experiments are too crude to make such a connection here.*

Response: In our original submission, we proposed “the cytoplasmic flow could also be driven by a gradient of hydrostatic pressure generated by the higher rachis contractility at the distal region compared to proximal”. This was a hypothesis and it had not been tested. In light of the reviewer’s comment we have removed this speculative sentence from our revised manuscript to avoid confusion for the readers.

The purpose of the cytoplasmic flow experiments (new figure 5) was to test whether the changes in cell shape that we observed under mild perturbation of actomyosin contractility regulators might be an indirect result of changes in cytoplasmic streaming. Importantly, our measurement show that the changes in cell shape are independent of changes in cytoplasmic streaming, since partial depletion of actomyosin regulators affects the cell shape without affecting flow. We have explained this in detail in our revised manuscript.

One way to block streaming that was previously used by the Wolfe/Priess study was to inject a drop of oil in the rachis. This experiment revealed that flows were still active proximal to the injection site but not distal. Perhaps this simple condition to block the cytoplasmic streaming and decouple distal and proximal streaming could be used to further monitor tension within the rachis (using laser cutting) and better link changes in rachis tension with hydrostatic pressure.

Response: We appreciate the reviewer’s suggestion to check the tension (by laser ablation) after oil injection into the gonad to stop the flow in the distal region. However, technically such an experiment is not feasible since injecting the gonad will physically affect the gonad and release the membrane tension at the injection site making the interpretation after laser ablation ambiguous. Furthermore, as explained above, we already provide evidence that germ cell shape responds to changes in tension that have no effect on cytoplasmic streaming, so we do not think there is a need to block cytoplasmic streaming when measuring tension with laser ablation.

Without this I am not sure what is the significance of the results reported in Figure S3.

Response: The significance of this experiment (now presented in figure 5), beyond the point explained in the previous response is also to show that actomyosin contractility *is* essential for cytoplasmic streaming. While weak perturbations of contractility regulators did not perturb cytoplasmic streaming, strong knock down of CYK-1, NMY-2 and MEL-11 resulted in a significant reduction of average speed and velocity of cytoplasmic particles (Figure 5). The previous study from Wolke et al (*Development* **134**, 2227-2236 (2007)) observed a slow rate of cytoplasmic flow after treatment with drugs which perturb contractility (actin-depolymerizing drug Latrunculin A and myosin light

chain kinase inhibitor- ML7) However, they did not report the role of any molecular regulator for this process. In our current study, we identified the role of formin-homology protein, CYK-1, non-muscle myosin-II (NMY-2) and myosin phosphatase (MEL-11) in regulating cytoplasmic streaming which reinforces the previous observation made by using drug manipulations.

Also, the finding by Wolfe et al. that the force generating streaming is in the proximal region is difficult to reconcile with the authors' finding that there is more tension in the distal region compared to the proximal region. This should be addressed.

Response: Wolke et al reported that cytoplasmic streaming is faster in the proximal region compared with the distal region. In their study, Wolke et al suggested that force for cytoplasmic streaming is generated near or within the oocytes present in the proximal region. The force-generating machinery for cytoplasmic streaming is not known, and it does not seem likely to be from contractility of the actomyosin corset. Therefore, we do not see any contradiction between our results and Wolke et. al.

We can speculate that faster flow rate in the proximal region is due to actomyosin forces generated within the oocytes, but since loss of contractility regulators would likely affect both the rachis tension and tension within the oocytes, we cannot dissect the role of the two tensions in the cytoplasmic streaming.

We note that our laser ablation experiments showing that there is more tension in the distal region compared with the proximal region of the rachis surface are consistent with previous immunostaining results of phosphorylated myosin in the *C. elegans* gonad (Nadarajan, S., Govindan, J.A., McGovern, M., Hubbard, E.J. & Greenstein, D. MSP and GLP-1/Notch signaling coordinately regulate actomyosin-dependent cytoplasmic streaming and oocyte growth in *C. elegans*. *Development* **136**, 2223-2234 (2009)). However, since the significance of the differential rachis tension in distal versus proximal regions is currently unclear we only included this observation in the supplementary section. We hope to explain the significance of this result in future studies.

3. I do not understand the significance of the FRAP analyses performed in Figure S1. While the experiments are well done, what is the significance of different actomyosin contractility components having different turnover rates at rachis bridges? How does this impact force generation within the rachis? Perhaps the authors could incorporate this information in their in silico model and use it to make predictions that can be experimentally tested? As written I do not understand what this brings to the story and/or what I should conclude from these experiments other than the results stated.

Response: The FRAP experiments are part of our detailed molecular characterization of the actomyosin corset, which we describe here for the first time. We don't know the significance of the differences in turnover rates between the different contractility components tested, but we still think they

should be reported. The more important point we make is that their turnover rates are significantly slower than have previously been measured for the same proteins in other actomyosin structures such as the zygote cortex or cytokinetic ring. Thus, our FRAP analysis in Supplementary Figure 1 shows that the actomyosin regulators present at the rachis are largely immobile and have low turnover indicating that they form a stable structure.

The role of turnover of these components on the force generation within the rachis could not be tested using our model because in our numerical model, contractility is implemented using the apical boundary contractility and this is a coarse-grained description accounting for the different contractile elements. Long time stable shapes of the germline are obtained from the simulations based on the prescribed levels of actomyosin contractility along the rachis edges. Whilst, to test a dynamical scenario as suggested will require a more complex model where contractility is implemented by specifying turnover rates at the edges of the apical rachis and using some formalism to implement tensions based on the polymerization and depolymerization rates. However, such a model is beyond the scope of this current work.

4. I do not have proper mathematical or computational expertise to assess whether or not the in silico model is correct. However, while I appreciate that the model recapitulates some of the observations that were made, I find it strange that the model is not tested per se. In my opinion, a model proves correct when it makes a prediction that can be experimentally verified, which is not the case here. In this sense I think that the model could be better exploited, for instance using the suggestions above.

Response: We thank the reviewer for this comment. It helped us realize that we should better connect our model with the experimental results and emphasize its main predictions. In particular, our model is consistent with experiments for the three observed phenotypes (WT, loss-of-contractility mutants, and excessive-contractility mutants) and thus validates that tensions alone can regulate the global shape of the germline. Furthermore, it predicts that at the given geometry of the system and imposed external constraints, the rachis cannot become modulated under spatially uniform apical contractility as would be expected based on previous works reporting buckling of tissues due to apico-basal asymmetry (Storgel, N., Krajnc, M., Mrak, P., Strus, J. & Zihlerl, P. Quantitative Morphology of Epithelial Folds. *Biophys J* **110**, 269-277 (2016); Krajnc, M. & Zihlerl, P. Theory of epithelial elasticity. *Phys Rev E Stat Nonlin Soft Matter Phys* **92**, 052713 (2015)). The model also shows that the shape can be modulated provided that the apical contractility varies. This is confirmed by the pearling-like phenotype observed in the *plst-1* mutant where the contractility does vary (new figure 6).

To further elaborate the irregularity of the germline shape, we have added two new simulations in which we test in our model mechanics-related contributions of cell shape and packing and contractility irregularity (Supplementary Fig. 6 and Sec. 3.2 of Supplementary text).

Furthermore, to improve quantitative agreement with experiments, we made a minor adjustment in our model. We now describe apical contractility by an energy term, which is quadratic in the apical perimeter; following analogous 2D models (Farhadifar, R., Roper, J.C., Aigouy, B., Eaton, S. & Julicher, F. The influence of cell mechanics, cell-cell interactions, and proliferation on epithelial packing. *Curr Biol* **17**, 2095-2104 (2007); Bi, D., Lopez, J., Schwarz, J. & Manning, M.L. A density-independent rigidity transition in biological tissues. *Nature Physics* **11**, 1074 (2015)); in the initial manuscript the relation was linear.

We moved the parametric phase diagram and the dependence of the rachis diameter on the apical contractility from Supplemental Material to the main text (Fig. 7c,f). Using the experimental findings presented in our manuscript (particularly those related to rachis diameter), we related these results to the observed phenotypes (dashed lines in Fig. 7c, f). Finally, we point to the importance of external constraints in maintaining the integrity (and shape) of the epithelial structure (Sec. 1 of Supplementary text). To elaborate this, we imaged the basement membrane with the germline. Assumption for our model with basal side being fixed is consistent with the experimental observation (Supplementary Fig. 4c). Effects of external constraints are often overlooked, which is why we find this result particularly important.

Finally, the 3D vertex model presented here is one of only a few 3D models that can relate intracellular processes with both cell-shape changes and ultimately 3D tissue-scale deformations (Okuda, S., Inoue, Y., Eiraku, M., Adachi, T. & Sasai, Y. Vertex dynamics simulations of viscosity-dependent deformation during tissue morphogenesis. *Biomech Model Mechanobiol* **14**, 413-425 (2015); Misra, M., Audoly, B., Kevrekidis, I.G. & Shvartsman, S.Y. Shape Transformations of Epithelial Shells. *Biophys J* **110**, 1670-1678 (2016); Bielmeier, C. *et al.* Interface Contractility between Differently Fated Cells Drives Cell Elimination and Cyst Formation. *Curr Biol* **26**, 563-574 (2016)). We expect this model to be used to study even more complex epithelial transformations, e.g. the dynamics of early stages of embryonic development in various animals, and could also help with detailed engineering of tissues *in vitro* (e.g. organogenesis).

Also, in silico modeling of forces in the rachis was previously carried out (Coffman et al., Biophys. J. 2016). While the authors document discrepancies between their work and that published in this article (with NMY-1 localization and phenotype), which suggests that this other model may rest on false premises, I think that it would be important to comment on the comparison between their approach to model the syncytial germline and this one.

Response: We carefully looked into the model proposed by Coffman et al. The model itself is based on fundamental physics and as such, we believe, is correct. However, it is built on the experimental observations that we could not reproduce such as- localization and loss of function phenotype of NMY-1 (Supplementary fig. 2b-c).

As for comparison of the models: In our model, there is a tug-of-war between rachis bridges of adjacent cells that pulls them in opposite directions to keep

them open. Coffman et al. describe this tug-of-war at the level of individual cells by a competition between contractility of the ring and (opposing) tension of the cortex. Second, their model does not give any information about the three-dimensional structure of the gonad such as diameter of the rachis or the shape of individual cells.

Since the work by Coffman et al. addresses completely different questions than our model, we were not able to find a suitable way to make a detailed comparison of both approaches in the main text. However, we do mention how our description of the rachis bridges differs from that in the work by Coffman et al. in section 1 of the Supplementary text.

Minor comments are suggestions:

5. Line 44: The authors state that C. elegans has 2 U-shaped gonads. The convention is that C. elegans has a single gonad that forms 2 U-shaped gonad arms.

Response: We thank the reviewer for pointing out this error in the manuscript. We have corrected it in the revised version.

6. In the first paragraph of p. 5, the authors conclude that “more than half of the tension in the germline is myosin-dependent” based on laser-cutting experiments done in animals depleted of nmy-2. As stated, this conclusion can be misleading because there is an underlying assumption that some of the tension is myosin-independent, which is clearly not possible to conclude from these results. I suggest rewording this section to make sure that it is clear that nmy-2 is partially depleted and therefore this only highlights the fact that myosin is a significant contributor to the tension (it could even account for all of it).

Response: We completely agree with the reviewer’s suggestion and have revised the text clearly stating that most if not all the tension in the germline is myosin-dependent.

7. The authors refer to the “contractome” when discussing contractility proteins. I am not quite sure what is encompassed by the term “contractome”. All previous work have referred to contractility regulators or proteins when discussing about the proteins studied here. I would therefore keep nomenclature straight and use “contractility regulators”, so as to avoid building confusion within the field.

Response: The contractome is a term we coined previously (Zaidel-Bar, R., Zhenhuan, G. & Luxenburg, C. The contractome--a systems view of actomyosin contractility in non-muscle cells. J Cell Sci 128, 2209-2217 (2015).), which encompasses all proteins that regulate actomyosin contractility in non-muscle cells. We added a citation immediately following the use of the term in the revised paper so readers who are not familiar with the term will easily get acquainted with it.

8. The genotype of many of the strains listed in Table S1 is not correct, namely COP1481, COP1234, COP937 and UN1608. Please use proper nomenclature.

Response: We thank the reviewer for pointing out this error. We have listed the aforementioned genotypes with proper nomenclature in the revised manuscript (Supplementary Table 1).

Reviewer #2 (Remarks to the Author):

*What is the mechanical basis for the maintenance of syncytial architecture, an important and conserved aspect of the germline? The present study tackles this question by focusing on the role of actomyosin contractility in the maintenance of germline syncytial organization in *C. elegans*. The authors employ an interdisciplinary approach, combining live fluorescence confocal microscopy of the early meiotic region of a gonad under genetic, pharmacological and mechanical perturbations with computational cell-based modelling of the common rachis. Based on their results, the authors conclude that in the plane of the rachis, actomyosin contractility counteracts tension to control the correct germ cell opening into the rachis, while in an orthogonal direction, contractility counteracts hydrostatic pressure and membrane tension to maintain the correct germ cell heights. Overall, this work demonstrates the role of actomyosin contractility in maintaining syncytial architecture.*

As a modeller, I am not well placed to comment on the validity of the experimental techniques used, nor their biological significance. However, I found this manuscript to be well written, the study itself to be clearly motivated, and the findings interesting. While no quantitative predictions were generated by the computational model, this nevertheless represents an innovative and complementary approach to studying the mechanics underlying syncytial architecture. In my view, this work would be of interest to researchers working in other areas of epithelial mechanics and morphogenesis.

Response: We thank the reviewer for recognizing the validity and relevance of our work. With our computational approach, we wanted to identify the relevant physical mechanisms in maintaining integrity and shape of the rachis tube, and to find the minimal model that would qualitatively agree with experimental observations. Indeed, we were able to come up with a theoretical description based on only two physical parameters, which is able to recapitulate both the wild-type situation and all observed mutants.

I have the following technical queries on the main text:

p.9 "We propose the flow could also be driven by a gradient of hydrostatic pressure generated by the higher rachis contractility at the distal region compared to proximal" - Do the authors' simulations bear out this hypothesis? Presumably they could visualize the pressure in each cell in a relevant simulation (related to the

fourth contribution to the author's choice of energy functional in equation (S3) in the supplementary information)?

Response: Our hypothesis referred to the pressure difference between the distal and proximal region of the gonad and not the pressure inside cells. Currently, we do not have any experimental evidence to support this hypothesis and our model is based on the distal region of the gonad and is not equipped to address it (elaborated below). Therefore, we have removed this hypothesis from our revised manuscript to avoid confusion for the readers.

At the core of the model, individual cells are parameterized by positions of vertices, allowing us to address the shape of individual cells in detail. Within such an approach, the interaction of the tissue with the environment is typically taken into account by different constraints without going into details of the behavior of the surrounding medium itself. Therefore the description of the cytoplasmic streaming is not possible in our current model. Indeed, this would require a hydrodynamic description where the entire domain (not just the epithelium) would be described using hydrodynamics-related quantities such as the flow and the associated velocity and pressure profiles. Some previous studies, including the paper by *He et al. "Apical constriction drives tissue-scale hydrodynamic flow to mediate cell elongation", Nature 508, 392 (2014)*, have indicated that cytoplasmic flow could be also driven by cell shape changes, which possibly indicates the need for a computational tool which would jointly address both cell-/tissue-shape changes and fluid dynamics. Development of such a model is far beyond the scope of this paper.

Nevertheless, the Reviewer is completely correct in pointing us to measure the internal pressure. The volumetric/bulk energy term is related to the pressure in cells. This pressure depends on cell compressibility and its volume and can be calculated using the Murnaghan equation of state. We now explain this in Sec. 1 of Supplementary text. In most cases studied in our manuscript, cells have the same compressibility and (at least on average) the same volume. Therefore, pressure is the same in all cells. This changes in the mutant with modulated contractility and the pressure in cells also modulates accordingly as described by Murnaghan equation of state.

p.11 "The gonadal tube was modeled as a cylindrical epithelial sheet of closely packed cells with hexagonal bases and apices" - Is this geometric assumption reasonable, based on the authors' live imaging data? From eyeballing Figure 1 it seems that the majority of cells are indeed hexagonal, though presumably it would be straightforward to quantify this. In either case, an additional note clarifying whether this is based on the experimental data, or an assumption designed to simplify the model calculations, would be helpful (would the authors expect their simulations results to be affected at all if cells were not all hexagonal?).

Response: We thank the reviewer for this question. Indeed, not all cells are hexagons in the real system. Our simple geometrical model where cell shape is taken into account in a coarse-grained way, i.e. only considering proper scaling of cell's linear dimensions and disregarding prefactors (Section 3 of

Supplementary text), suggests that details regarding polygonality are not important, which is why we have not studied them in the initial version of the manuscript.

Nevertheless, our computational tool allows us to easily introduce disorder of cell packing by imposing random T1 transformations of the junctional polygonal network. We therefore repeated the simulations for the wild type and the mutant phenotypes starting with a disordered network. Indeed, we show in the revised manuscript that the overall (average) rachis diameter is not significantly affected in different conditions, however the detailed shape is somewhat less regular, i.e. cell heights vary a bit (Supplementary Fig. 6).

In addition, I have the following queries on the supplementary information for their 3D vertex model:

p.2: "We assume a first-order dynamics given by the overdamped equation of motion for vertices" - As noted by the authors just after equation (S5), the assumption is made that all vertices have the same friction drag coefficient. Since this term is proportional to the velocity of each vertex, it would seem to represent friction against the underlying matrix. Is there extracellular matrix or a basement membrane either apical or basal to the germ cells? If so, is this uniform in space; if not, could a cell-cell friction term be more suitable?

Response: Our model dynamics describes cell-shape changes and cell movements implicitly through displacements of vertices. We assume this dynamics to be overdamped and assume only friction with the environment. This is the simplest description used in all vertex models. Indeed, at the basal side, cells are attached to the basement membrane (as shown by our new results in Supplementary Fig. 4c), and the interior of the rachis tube is filled with the common cytoplasm i.e., a viscous fluid. Adding cell-cell friction might be important for cell dynamics, e.g. tissue rheology, which we do not address here. Since, we are interested in equilibrium configurations of tubes and therefore overdamped dynamics is employed as a numerical procedure for energy minimization (i.e. steepest descent method).

p.4 "The basal side of the gonadal tube is fixed" - Is this assumption based on the authors' experimental observations? Given their findings concerning the lack of buckling/undulations in simulations under differences in apico-basal tension, I am wondering if this behaviour might occur if the basal surface were allowed to deform to some extent?

Response: Indeed, at the basal side, cells are attached to the basement membrane, made of laminin. We imaged the basement membrane to validate this assumption (Supplementary Fig. 4c).

One of the co-authors of our manuscript previously explored buckling due to apico-basal asymmetry in a simplified 2D case (Štorgel, N., Krajnc, M., Mrak, P., Štrus, J. & Zihelr, P. Quantitative Morphology of Epithelial Folds. *Biophys J* **110**, 269-277 (2016); Krajnc, M. & Zihelr, P. Theory of epithelial elasticity. *Phys Rev E*

92, 052713 (2015)). This study shows that the apico-basal differential surface tension changes sign of the effective bending modulus of the tissue. This leads to a buckling instability, which collapses the tissue, preferring infinite local curvature. There, the collapse is prevented by the interaction with the substrate or the extracellular matrix and by the steric repulsion between cells. Based on the 2D case, we expect that allowing the basal side of the tissue to deform while keeping all other conditions the same would not be sufficient to observe buckling. In addition, one would need to also allow the length of the tube to change and include either steric repulsion between non-neighboring cells or bending rigidity of the basement membrane (or both) so as to prevent total collapse of the tissue. The length of the tube would then need to be varied so as to minimize the energy. Therefore, to answer the reviewer's question, we performed a trivial simulation, in which we released the basal vertices from the constraint while keeping the fixed length of the tube. This caused both the apical and the basal diameter to decrease and no buckling was observed. Please refer to the figure below:

Figure. Both the rachis and the gonad diameter decrease due to lateral surface tension and apical contractility after releasing the basal side of the tube from constraint.

Again, allowing variation of the tube's length would lead to buckling (as observed in arXiv:1805.06500 [cond-mat.soft]). We believe that these results go beyond the current work and so we chose not to add them to the revised manuscript. Nevertheless, we did elaborate on the importance of constraints (Sec 1 of Supplementary text).

Finally, I spotted the following minor typographical errors:

p.8: "in two orthogonal direction" -> "in two orthogonal directions"

p.17: "The germ cells enter mitosis at the distal end and progresses" -> "The germ cells enter mitosis at the distal end and progress"

p.17: "to form mature oocyte" -> "to form a mature oocyte" or "to form mature oocytes"

Methods: "Height of the germ cells" -> "The germ cell heights"

Supplementary Video Legends: "Green hexagons represents apical end of the germ cells" -> "Green hexagons represent germ cells' apical ends"

Response: We thank the reviewer for pointing out these errors. We have made all the corrections in the revised manuscript.

Reviewer #3 (Remarks to the Author):

This manuscript focuses on the role of actomyosin contractility in maintaining the organization of the syncytial gonad of C. elegans. The authors do a very nice job in describing the structure of a corset-like sheath composed of Actin, Myosin and several of their regulators that separates forming oocytes from a central, tubular shaft or rachis. Rachis bridges form a direct connection between the oocytes and the rachis and are somewhat similar to the ring canals that connect germ cells in the Drosophila ovary. Components of an actomyosin "contractome" that co-localize with actin and myosin in the sheath include anillin, the actin bundling protein PLST-1, a septin, a RhoGAP. The authors used FRAP analysis to evaluate protein stability in the corset and find that different components recover from photobleaching at different rates and have different characteristic mobile fractions. Nevertheless, all proteins tested exchanged more slowly than from the cortex of cells, suggesting that the rachis sheet is particularly stable, with slow turnover and large, immobile pools. In addition, the authors used laser ablations to show that the actomyosin corset is under tension, both circumferentially and longitudinally (however, see specific comments below). Moreover, they show that various strategies designed to reduce the level of actomyosin contractility (e.g., myosin knockdown or the use of a temperature sensitive formin allele) reduce sheath tension. Further analysis of reducing actomyosin contractility through depletion of NMY-2 (but not NMY-1) could, at extreme levels cause sterility, but at intermediate levels caused the height of the germ cells to decrease (because the sheath was less contractile) and caused the diameter of the rachis bridges to increase. Similar results were caused by compromising the activity of the Rho kinase, Let-502 or the formin, cyk-1 and pharmacological strategies to compromise actomyosin function also perturbed corset structure. In contrast, depletion of negative regulators of actomyosin function had the opposite effect (see query in the specific comments below). The authors further show that cytoplasmic flow is reduced when actomyosin contractility is compromised and suggest that the flow is driven by a hydrostatic pressure gradient caused by increased rachis contractility in the distal region of the gonad. A final series of experiments focused on the morphology of gonads from worms defective in PLST-1, a plastin homolog that assures a smooth distribution of actin and myosin, and actomyosin based tension in the rachis and rachis bridges. Finally, the authors generate an in-silico model for germline architecture based on a 3D vertex approach and show the model recapitulates the steady state shape of the gonadal tube. Moreover, increasing or

decreasing in silico tension recapitulates changes in rachis diameter, consistent with experiment. Introduction of a tube with apical tensions that vary periodically further mimic the effects of PLST-1 mutants.

Overall, the experiments and the model confirm the importance of the contractile actomyosin sheath in maintaining gonad structure. Ultimately, the paper should make a fine contribution to Nature Communications. Nevertheless, the specific comments below should be addressed before it can be published.

Specific comments.

Lines 94-105. The authors report an initial recoil velocity for a variety of conditions, but do not report the time-course of recoil. Was distance of recoil vs. time linear? Exponential? A sum of two exponentials? In the supplement, the authors state that "Initial recoil velocity was obtained by calculating the average recoil velocity in the first ten seconds after the ablation." This only makes sense if the recoil was linear with time and averaging was designed to account for noise. It might possibly be a very poor measure of initial recoil velocity if recoil decay is exponential or the sum of exponentials. The authors need to tell the reader what the recoil looked like (present distance vs. time in a supplemental figure) and convince the reader that their method of calculating an average is appropriate.

Response: We thank the reviewer for pointing out the importance of showing the entire time-course of recoil and calculating the initial recoil from only the very first time points. We have recalculated the initial recoil velocity by using the first two seconds just after the ablation to avoid any contribution caused due to the viscoelastic property of the tissue. Initial recoil velocity is nearly linear during this period of time (Red dotted line shown in Fig. 2c and 2h). We have revised the method section explaining the quantification of initial recoil velocity. Also, we have included a time-course of displacement of the membrane after laser ablation in the revised manuscript (Fig. 2c and 2h).

The 8 higher magnification panels in Figure 1g do not have an associated scale bar and one should be added.

Response: We have added the scale bar in the revised figures as suggested.

Line 107. The authors state "Next, we sought to decrease the level of actomyosin contractility and observe the effects on syncytial germline structure." This statement is somewhat confusing given that line 96 deals with the depletion of myosin activity and lines 99-103 deals the effect on recoil of natural changes in the level of myosin phosphorylation and therefore activity.

Response: We have rephrased the sentence to "Next, we sought to observe the effects of decreasing the level of actomyosin contractility on syncytial germline structure."

Line 112. Why do the authors introduce *cyk-1* here, but wait until line 490 to tell us it's a formin?

Response: We thank the reviewer for pointing this out. In the revised manuscript, we mention CYK-1 as being a formin when it is first mentioned.

Line 185. The authors need to explain why "excessive contractility", which, according to line 168 causes a small but significant increase in the height of the germ cells would end up causing small, non functional oocytes. Shouldn't a smaller rachis and taller germ cells give rise to taller oocytes? Perhaps the authors need to explain better what happens in the proximal region as the oocytes transition through the U turn that the gonad makes? This comes up again on line 291-292.

Response: To explain why excessive contractility leads to formation of small and non-functional oocytes, we have included a new figure focusing on the loop region (U-turn) of the control and mutants with excessive contractility along with better explanation in the revised manuscript (Supplementary Fig. 3). Oocyte growth predominantly depends on the cytoplasmic flow from the distal rachis into the proximal germ cells (Wolke, U., Jezuit, E.A. & Priess, J.R. Actin-dependent cytoplasmic streaming in *C. elegans* oogenesis. *Development* **134**, 2227-2236 (2007); Kim, S., Spike, C. & Greenstein, D. Control of oocyte growth and meiotic maturation in *Caenorhabditis elegans*. *Adv Exp Med Biol* **757**, 277-320 (2013)). Although, the germ cells in the mutants with excessive contractility are taller, they have extremely narrow germ cell opening (rachis bridge) as compared to control (Fig. 4a and 4c) and have hyper-constricted rachis in the distal region (Fig. 4a and 4d) as well as in the loop region (Supplementary Fig. 3), which probably results in insufficient transfer of the cytoplasm leading to the formation of small-sized oocytes.

Also, the rate of cytoplasmic flow is significantly slow in the *mel-11*(RNAi) worms as compared to the control (Figure 5, Average speed and average velocity in wild-type: $1.50 \pm 0.50 \mu\text{m}/\text{min}$.; $4.71 \pm 1.63 \mu\text{m}/\text{min}$ and *mel-11*(RNAi) worms: $0.45 \pm 0.80 \mu\text{m}/\text{min}$; $3.09 \pm 1.26 \mu\text{m}/\text{min}$, respectively).

Finally, the *in silico* model, which is detailed in the Supplement is likely to be attractive to many mathematicians interested in morphogenesis. It could be presented in a much more useful fashion and should be. Specifically, the model appears to fit well the biology being studied. Nevertheless, like all models it requires simplifications. The authors are best suited to explain which simplifications fit the biology almost precisely, and which fall short and how. Further, the model as written (like many such models) seems to be written for experts in modeling. The description of the model in the main text is fine. Both additions to the model, a more biologically friendly description of the model and a careful analysis of how each of the assumptions that allow simplification maps to the biology could come simply by interspersing each section devoted to the model in the supplementary material with appropriate expansions of the supplementary text, and where useful, additional supplementary figures.

Response: We thank the reviewer for recognizing the validity of our model. We agree that the description of our approach in Supplemental Material of the initial

manuscript was too technical. Following the reviewer's comment, we rewrote the Supplemental Material in a more compact way as described below. In addition, we also rewrote the description of the model in the main text results section. The description there explains what the different parameters of the model are and the assumptions made in building the model. We also added to the main figure two phase diagrams that explore parameter space for the salient components of the model (Figure 7).

The modeling part of Supplemental Material of the revised manuscript now contains 3 sections. In Section 1, we describe only the most essential ingredients of our approach, introducing the model parameters and citing the relevant literature. In this section we also describe the main assumptions and simplifications. Assumptions pertaining to the vertex dynamics and model energy functional [Eqs. (1) and (2)] are standard in vertex models, whereas assumptions related to boundary conditions are system-specific and are in fact a crucial part of our analysis. We describe these assumptions in more detail in the last paragraph of Sec. 1 of the revised modeling part of Supplemental Material. To justify them, we performed additional experiments, where we imaged the basement membrane (Supplementary Fig. 4c). We believe that Section 1 of the modeling part of Supplementary text should now be easier to read (both for biophysicists and biologists). For the modeling community that would be interested in implementing similar models, we give a more detailed description of the implementation in Section 2. The content of this Section gives the minimal amount of information needed by an experienced computational person to be able to reproduce our results. Of course, implementation of our model, i.e. to write the computer code for it, is quite involved and therefore, we will soon share our tool via an open source license together with a more technical manuscript focusing only on the computational method. Finally, in Sec. 3, we apply the model to our experimental system (assuming experimentally measured geometric parameters, e.g. diameter of the gonad, length of the tube, etc.) and present all the results.

REVIEWERS' COMMENTS:

Reviewer #1 (Remarks to the Author):

The revised manuscript by Priti et al. is significantly improved from the previous version and I consider that most of the concerns raised by the reviewers have been well addressed and clarified. I still have a few minor points that the authors should consider before publication.

1) In line 59-60 and Figure 1C, the authors describe what the germline looks like in an orthogonal view and indicate that there are 10-12 germ cells surrounding the structure. Obviously this number of germ cells is characteristic of a specific region of the gonad, as there should perhaps be fewer germ cells in the very distal and proximal ends. Perhaps the authors should consider adding to this sentence which region of the germline was used to make this section, so that it is easier to contextualize the observation.

2) I do not think that my previous comment regarding the inappropriate genotype of the strains in Table S1 has been addressed properly, despite a statement from the authors that it was. A designation such as "cyk-4(knu286 - C-terminal eGFP LoxP)" is technically inaccurate and lends to confusion. I strongly advise the authors to consult the reference for *C. elegans* nomenclature on Wormbase (https://wormbase.org/about/userguide/nomenclature#6gm98kaelihf7b_0315d42c--10) and pay attention to the section on genome engineering so that the genotype of strains expressing tagged genes is properly designated. In the long run this will help the authors by alleviating confusion on the reagents that they have used. All genotypes in Table S1 should be checked to make sure that they conform to *C. elegans* accepted conventions (and specifically the genotypes for strains COP1481, COP1234, COP937 and UN1608, all of which are obviously incorrect).

Reviewer #2 (Remarks to the Author):

The authors have thoroughly addressed my technical queries regarding the modelling component of their manuscript.

Reviewer #3 (Remarks to the Author):

Priti et al.

The authors have now addressed most of my concerns about the experimental part of the work adequately and the experimental part of the manuscript requires only minor modifications for publication. The model is now much easier to understand and I think it contributes nicely to the overall work presented. Nevertheless, I am very confused about cell compressibility and the authors really need to better describe what they mean by compressibility and where it comes from. See my comments below.

I think that they should address the following, minor comments on the experimental piece:

Line 40. They should define rachis with a clause or a sentence or two. It's obvious what it is from the figure, but a definition would help the reader.

Line 55. It is obvious that anillin is highly concentrated at "the top of the T-shaped radial partitions...". Nevertheless, to claim that anillin is "exclusively" at that site is beyond what the authors know for sure. It may seem picky, but anillin may have important, as yet unknown functions elsewhere where it is simply not detected using the imaging strategies they are using. Change to reflect "highly concentrated".

Lines 103 and 104. The authors responded to my previous comment that measuring rate by taking a point 10 secs out and calculating the average rate to get there does not constitute an "initial rate" by changing the interval to two seconds. Unfortunately, this still does not adequately

estimate an initial rate and indeed, careful inspection of the data suggest to me that their method over estimates the differences between control and experimental (I appreciate that there is a difference). Getting rates from discontinuous data is inherently difficult an the usual way of doing so is by differentiating smoothed data obtained by curve fitting. Presumably, their data could be fit by one or two exponentials. Another issue is where was the cut actually made compared to the interval at which images were recorded. Peralta et al 2007 (Biophys J. 92: 2583, see Appendix C under "Determination of the recoil velocity") describes how to deal with noisy data and uncertainty in the time of the cut. Should the authors prefer not to use this approach, at least they should be clear that they are not really looking at initial velocity but instead an average velocity taken by looking at displacement after 2 seconds.

Line 109. Start a new paragraph at "To test the presence...". This sentence starts on what can be viewed as a new set of experiments and the paragraph will help the reader.

The more significant issue they need to address is on the model:

Line 298. The authors need to explain where what they are calling cell compressibility. It is generally accepted that at biologically relevant pressures, water is essentially incompressible. Changes in volume due to compressibility is therefore taken as flow of water or water and solute out or into the cell. Is that what the authors mean?

Finally. The authors need to review the Figure legends carefully. For example, I did not see an adequate description of supplementary figure 4b. If everything in the figure is properly described in the text or in the supplement (my apologies if I missed it), at least say so in the legend so the reader can easily find the information.

Point by point response to Reviewers' comments:

Reviewer #1 (Remarks to the Author):

The revised manuscript by Priti et al. is significantly improved from the previous version and I consider that most of the concerns raised by the reviewers have been well addressed and clarified. I still have a few minor points that the authors should consider before publication.

1) In line 59-60 and Figure 1C, the authors describe what the germline looks like in an orthogonal view and indicate that there are 10-12 germ cells surrounding the structure. Obviously this number of germ cells is characteristic of a specific region of the gonad, as there should perhaps be fewer germ cells in the very distal and proximal ends. Perhaps the authors should consider adding to this sentence which region of the germline was used to make this section, so that it is easier to contextualize the observation.

Response: In the orthogonal view, we observed 10-12 germ cells surrounding the rachis in the early meiotic region of the gonad (~80-100 µm from the distal end). We have included this information in our revised manuscript.

2) I do not think that my previous comment regarding the inappropriate genotype of the strains in Table S1 has been addressed properly, despite a statement from the authors that it was. A designation such as “cyk-4(knu286 - C-terminal eGFP LoxP)” is technically inaccurate and lends to confusion. I strongly advise the authors to consult the reference for *C. elegans* nomenclature on Wormbase (<https://wormbase.org/about/userguide/nomenclature#6gm98kaelihf7b0315d42cj--10>) and pay attention to the section on genome engineering so that the genotype of strains expressing tagged genes is properly designated. In the long run this will help the authors by alleviating confusion on the reagents that they have used. All genotypes in Table S1 should be checked to make sure that they conform to *C. elegans* accepted conventions (and specifically the genotypes for strains COP1481, COP1234, COP937 and UN1608, all of which are obviously incorrect).

Response: We thank the reviewer for pointing out this error. As suggested by the reviewer, we consulted the reference for *C. elegans* nomenclature on Wormbase and listed the aforementioned genotypes with proper nomenclature in the revised manuscript.

These are the corrected genotypes:

COP1481: *unc-119(ed3) III; unc-59 (knu463 [unc-59::degron::mKate2 + loxP unc-119 (+) loxP]) I*

COP1234: *cyk-4 (knu286 [cyk-4::GFP + loxP]) III*

COP937: *unc-119(ed3) III; cyk-1 (knu84 [cyk-1::GFP + loxP unc-119 (+) loxP]) III*

UN1608: *nmy-1(xb5[nmy-1::mKate2]) X*

Reviewer #2 (Remarks to the Author):

The authors have thoroughly addressed my technical queries regarding the modelling component of their manuscript.

Response: We are thankful to the reviewer for the positive comments about our manuscript.

Reviewer #3 (Remarks to the Author):

The authors have now addressed most of my concerns about the experimental part of the work adequately and the experimental part of the manuscript requires only minor modifications for publication. The model is now much easier to understand and I think it contributes nicely to the overall work presented. Nevertheless, I am very confused about cell compressibility and the authors really need to better describe what they mean by compressibility and where it comes from. See my comments below.

I think that they should address the following, minor comments on the experimental piece:

Line 40. They should define rachis with a clause or a sentence or two. It's obvious what it is from the figure, but a definition would help the reader.

Response: We have added the following sentence in the introduction to define the rachis: "The germ cells are arranged peripherally within the gonad and are only partially enclosed with plasma membrane, sharing a common central cytoplasm known as rachis."

Line 55. It is obvious that anillin is highly concentrated at "the top of the T-shaped radial partitions...". Nevertheless, to claim that anillin is "exclusively" at that site is beyond what the authors know for sure. It may seem picky, but anillin may have important, as yet unknown functions elsewhere where it is simply not detected using the imaging strategies they are using. Change to reflect "highly concentrated".

Response: We agree with the reviewer's comment about the localization of Anillin. Therefore, we have revised the text as follows: "Consistent with those reports, single medial plane views showed the presence of high concentration of ANI-2 at the top of T-shaped radial partitions composed of the germ cell membranes surrounding the central rachis".

Lines 103 and 104. The authors responded to my previous comment that measuring rate by taking a point 10 secs out and calculating the average rate to get there does not constitute an "initial rate" by changing the interval to two

seconds. Unfortunately, this still does not adequately estimate an initial rate and indeed, careful inspection of the data suggest to me that their method over estimates the differences between control and experimental (I appreciate that there is a difference). Getting rates from discontinuous data is inherently difficult and the usual way of doing so is by differentiating smoothed data obtained by curve fitting. Presumably, their data could be fit by one or two exponentials. Another issue is where was the cut actually made compared to the interval at which images were recorded. Peralta et al 2007 (Biophys J. 92: 2583, see Appendix C under “Determination of the recoil velocity”) describes how to deal with noisy data and uncertainty in the time of the cut. Should the authors prefer not to use this approach, at least they should be clear that they are not really looking at initial velocity but instead an average velocity taken by looking at displacement after 2 seconds.

Response: We thank the reviewer for suggesting a more accurate way of measuring recoil velocity. As suggested in our previous response, we have recalculated the initial recoil velocity as follows:

We tracked manually distance between two nearby membrane edges at each time point using MtrackJ plugin of the Fiji. Before the laser ablation, distance between two edges was fitted using linear function. After ablation, we used double exponential function to fit the data. We used single exponential function in cases where the recoil velocity was very less and double exponential function overestimated the velocity. The effective time of laser cut was estimated as the intersection of linear and exponential fitting curve. The recoil velocity was calculated, using Matlab software, as a derivative of the exponential function at the effective time of laser cut (Peralta, X.G. *et al.* Upregulation of forces and morphogenic asymmetries in dorsal closure during Drosophila development. *Biophys J* **92**, 2583-2596 (2007); Hara, Y., Shagirov, M. & Toyama, Y. Cell Boundary Elongation by Non-autonomous Contractility in Cell Oscillation. *Curr Biol* **26**, 2388-2396 (2016)).

We have included this information in the Methods section.

Line 109. Start a new paragraph at “To test the presence...”. This sentence starts on what can be viewed as a new set of experiments and the paragraph will help the reader.

Response: We agree with the reviewer’s comment and have revised the manuscript as suggested.

The more significant issue they need to address is on the model:

Line 298. The authors need to explain where what they are calling cell compressibility. It is generally accepted that at biologically relevant pressures, water is essentially incompressible. Changes in volume due to compressibility is therefore taken as flow of water or water and solute out or into the cell. Is that what the authors mean?

Response: The description of the cell compressibility term is based on Murnaghan equation of state used to describe simple fluids. [(Riest, J., Athanasopoulou, L., Egorov, S.A., Likos, C.N. & Ziherl, P. Elasticity of polymeric nanocolloidal particles. *Sci Rep* **5**, 15854 (2015)]. This description effectively captures cell-volume changes due to compressibility of intracellular cytoplasmic fluid, cortical network present within the cell as well as passive exchange of material with the environment, as indicated by the reviewer. Although, we do not explicitly model any osmotic or active transport due to flux of solutes or fluid in our numerical model as it is beyond the scope of current numerics, the phenomenological description in terms of the free energy has proven sufficient to study mechanics of cell- and tissue shape changes. A more refined model would include additional terms for passive and active fluid/solute fluxes as recently proposed by one of the authors [Dasgupta, S., Gupta, K., Zhang, Y., Viasnoff, V. & Prost, J. Physics of lumen growth. *Proc Natl Acad Sci U S A* **115**, E4751-E4757 (2018)] for simple cell aggregates.

We have included a clear description of compressibility in the revised manuscript.

Finally. The authors need to review the Figure legends carefully. For example, I did not see an adequate description of supplementary figure 4b. If everything in the figure is properly described in the text or in the supplement (my apologies if I missed it), at least say so in the legend so the reader can easily find the information.

Response: We thank the reviewer for pointing out this error. We have carefully gone through all figure legends in the revised manuscript and revised them appropriately.